# Sex-dependent effects of in utero cannabinoid exposure on cortical function

**Anissa Bara**[1,2†], **Antonia Manduca**[1,2,3†], **Axel Bernabeu**[1,2,4], **Milene Borsoi**[1,2], **Michela Serviado**[3], **Olivier Lassalle**[1,2], **Michelle Murphy**[2,5,6], **Jim Wager-Miller**[2,5,6], **Ken Mackie**[2,5,6], **Anne-Laure Pelissier-Alicot**[1,2,4,7], **Viviana Trezza**[3], **Olivier J Manzoni**[1,2]*

[1]Aix Marseille University, INSERM, INMED, Marseille, France; [2]Cannalab, Cannabinoids Neuroscience Research International Associated Laboratory, Indiana University, Indiana, United States; [3]Section of Biomedical Sciences and Technologies, Department of Science, University Roma Tre, Rome, Italy; [4]APHM, CHU Conception, Service de Psychiatrie, Marseille, France; [5]Department of Psychological and Brain Sciences, Indiana University, Bloomington, United States; [6]Gill Centre, Indiana University, Bloomington, United States; [7]APHM, CHU Timone Adultes, Service de Médecine Légale, Marseille, France

*For correspondence:
olivier.manzoni@inserm.fr

†These authors contributed equally to this work

**Abstract** Cannabinoids can cross the placenta, thus may interfere with fetal endocannabinoid signaling during neurodevelopment, causing long-lasting deficits. Despite increasing reports of cannabis consumption during pregnancy, the protracted consequences of prenatal cannabinoid exposure (PCE) remain incompletely understood. Here, we report sex-specific differences in behavioral and neuronal deficits in the adult progeny of rat dams exposed to low doses of cannabinoids during gestation. In males, PCE reduced social interaction, ablated endocannabinoid long-term depression (LTD) and heightened excitability of prefrontal cortex pyramidal neurons, while females were spared. Group 1 mGluR and endocannabinoid signaling regulate emotional behavior and synaptic plasticity. Notably, sex-differences following PCE included levels of mGluR1/5 and TRPV1R mRNA. Finally, positive allosteric modulation of mGlu5 and enhancement of anandamide levels restored LTD and social interaction in PCE adult males. Together, these results highlight marked sexual differences in the effects of PCE and introduce strategies for reversing detrimental effects of PCE.
DOI: https://doi.org/10.7554/eLife.36234.001

## Introduction

With 181.8 million estimated regular users aged 15 – 64, cannabis is the most commonly consumed illicit drug around the world (UNODC, 2015). The majority of users are young, and exposure largely occurs during their reproductive years (SAMSHA, 2015). In fact, data collected from 2002 to 2014 in the U.S. indicates that 7.5% of pregnant women between 18 and 25 years of age use cannabis, while the rate of use in all pregnant women is approximately 4% (*Brown et al., 2017*). A longitudinal prospective study on drug use in pregnancy showed that 48% of women who used cannabis in the year prior to their pregnancy continued to smoke cannabis throughout pregnancy (*Moore et al., 2010*). Moreover, 70% of pregnant and non-pregnant women believe that there is little or no harm in using cannabis once or twice per week (*Ko et al., 2015*). Thus, cannabis consumption during pregnancy is a major public health concern worthy of study as thousands of infants are prenatally exposed to cannabis.

Worldwide legalization and/or decriminalization initiatives emphasize the urgent need for scientific evidence regarding possible harms and benefits of cannabis use. Epidemiological studies of children with a history of in utero cannabis exposure reveal normal intelligence scores but altered cognition and executive functions (attention, memory, concentration, inhibitory control) and higher levels of depression and anxiety during adolescence (*Grant et al., 2018*; *Higuera-Matas et al., 2015*) have been reported. In animal studies, prenatal exposure to various regimens of cannabinoids has been linked to alterations in locomotor, learning, memory, emotional functions and social interaction (*Campolongo et al., 2011*; *Higuera-Matas et al., 2015*; *Vargish et al., 2017*). A major caveat is that most studies investigated effects using male progeny only.

Δ9-Tetrahydrocannabinol (THC), the major psychoactive ingredient in marijuana and other cannabis-based preparations, efficiently crosses the placental barrier and reaches the fetus (*Hurd et al., 2005*; *Hutchings et al., 1989*). In the brain, cannabinoid receptor type 1 (CB1R) is the main target of THC. The endocannabinoid system, mainly through CB1R, has been implicated in neuronal proliferation, migration, morphogenesis and synaptogenesis (*Berghuis et al., 2007*; *Harkany et al., 2007*; *Maccarrone et al., 2014*; *Mulder et al., 2008*). Consequently, it is crucial to understand the long-term consequences of cannabinoid exposure at this critical stage of development in both sexes.

Here, we examined how prenatal cannabinoid exposure (PCE) (to synthetic and plant-derived cannabinoids) influences the synaptic and behavioral functions of the medial prefrontal cortex, a brain region implicated in neuropsychiatric disorders. We observed marked sexual divergence in the protracted effects of PCE. Following PCE, social interaction was altered in adult male rats and this alteration was paralleled by sex-specific deficits in synaptic plasticity, neuronal excitability and relevant mRNA levels in the mPFC. We discovered that pharmacological potentiation of mGlu$_5$ or blockade of anandamide degradation both normalized behavioral and synaptic deficits. The data reveal new endophenotypes of PCE and identify new therapeutic strategies.

## Results

### Sex differences in the behavioral effects of prenatal cannabinoid exposure in the adult progeny

In rodent models, the consumption of synthetic or plant-derived cannabinoids during gestation has multiple deleterious consequences on the progeny's behavior (reviewed in *Campolongo et al., 2011*; *Higuera-Matas et al., 2015*; *Vargish et al., 2017*). In these earlier studies, different drug and treatment regimen were used and multiple behaviors studied in the progeny at various ages. Here, to simplify the analysis/maximize the significance of our comparisons of molecular, synaptic and behavioral measures we first decided to generate a coherent new dataset. Thus, gestating dams were exposed a single daily low dose of the synthetic cannabinoid WIN55,212 – 2 (WIN, 0.5 mg/kg, s.c. or vehicle or SHAM) between GD5 and GD20. Key findings were reproduced with the phytocannabinoid Δ9-Tetrahydrocannabinol, THC (5 mg/kg). This protocol of PCE is clinically relevant, since the doses of WIN and THC used here correspond to a moderate exposure of cannabis in humans, correcting for the differences in route of administration and body weight surface area (*Garcia et al., 1998*; *Mereu et al., 2003*). We compared male and female SHAM and cannabinoid-exposed progeny in a series of behavioral tests to explore social behavior, anxiety, locomotion and cognition at the adult stage (PND >90). In accord with international ethical guidelines to reduce the number of animal used and their treatment/manipulations, once we had established that there was no difference between sham and naive animals (*Table 1*), data from animals in both groups were pooled and naive rat dams were included in the SHAM group.

We found that PCE impaired specific components of social interactions in adult male rats. Thus, PCE males had less contact and spent less time interacting with their congeners than those exposed in-utero to vehicle (*Figure 1A–B*). Detailed exploration of the various parameters of social interaction revealed that sniffing (*Figure 1C–D*) and playing (*Figure 1E*) behaviors were impaired in PCE males while the number of attacks remained unchanged (*Figure 1F*). The low socialization in PCE males was unlikely due to impaired motor locomotion, as we found no significant change in the distance traveled between WIN and SHAM groups during testing (3758 ± 117.8 cm, n = 10; and 3779 ± 228.8 cm, n = 8; $t_{(7)}$ = 0.02, p>0.05, t-test; SHAM- and WIN-exposed rats, respectively).

**Table 1.** Statistical significance of different behavioral tasks in SHAM and Naive animals.

Prenatal treatment from GD 5 to GD 20 with vehicle solution in control dams (SHAM group) did not alter sociability (social interaction), anxiety (elevated plus maze), and cognition (temporal order and object recognition memory tasks) in adult rats of either sex. Statistical significance was determined using the unpaired Student's t-test after outliers were detected and removed from the dataset using Grubbs' test.

| Behavior | Parameter | Treatment | Sex | n | Mean | SEM | P value (Unpaired t test) |
|---|---|---|---|---|---|---|---|
| Social Interaction | # of contacts | NAIVE | Male | 5 | 216.200 | 28.420 | 0.439 |
| | | SHAM | | 10 | 235.800 | 10.550 | |
| | | NAIVE | Female | 4 | 211.800 | 21.600 | 0.85 |
| | | SHAM | | 7 | 215.100 | 5.713 | |
| Elevated Plus Maze | % Time Open | NAIVE | Male | 6 | 25.900 | 2.988 | 0.665 |
| | | SHAM | | 7 | 24.270 | 2.245 | |
| | | NAIVE | Female | 9 | 25.530 | 2.464 | 0.359 |
| | | SHAM | | 9 | 22.540 | 1.978 | |
| | % Open Entries | NAIVE | Male | 6 | 27.740 | 3.248 | 0.164 |
| | | SHAM | | 7 | 33.950 | 2.678 | |
| | | NAIVE | Female | 9 | 33.160 | 2.520 | 0.475 |
| | | SHAM | | 9 | 30.230 | 3.111 | |
| Temporal order | Discrimination index | NAIVE | Male | 8 | 0.320 | 0.085 | 0.702 |
| | | SHAM | | 7 | 0.366 | 0.078 | |
| | | NAIVE | Female | 8 | 0.258 | 0.132 | 0.896 |
| | | SHAM | | 7 | 0.277 | 0.038 | |
| Object Recognition | Discrimination Index | NAIVE | Male | 10 | 0.547 | 0.087 | 0.792 |
| | | SHAM | | 8 | 0.580 | 0.086 | |
| | | NAIVE | Female | 7 | 0.323 | 0.126 | 0.481 |
| | | SHAM | | 7 | 0.421 | 0.045 | |

DOI: https://doi.org/10.7554/eLife.36234.002

Interestingly, we discovered that the deleterious effects of PCE on social behavior were specific to male offspring. In PCE adult female rats, social behavior was indistinguishable from that of the SHAM group; there was no difference in the total frequency and time (*Figure 1*) of social interaction in either group.

Social behaviors require emotional control and cognitive abilities. To test for the generalization of PCE-induced deficits across these behavioral dimensions we estimated anxiety and cognition in our model. No differences between WIN- and SHAM prenatally-exposed rats were found in the elevated plus maze. Specifically, there was no change in the percentage of time spent in the open arms (*Figure 2A*), percentage of open arm entries (*Figure 2B*), in the time spent in the open and close arms (*Figure 2C–D*), number of closed-arm entries ([$F_{(treat \times sex)1,29}$ = 0.195 p=0.662]; data not shown), stretch-attended posture ([$F_{(treat \times sex)1,29}$ = 0.167 p=0.685]; data not shown) and frequency of head-dippings ([$F_{(treat \times sex)1,29}$ = 0.338 p=0.565; data not shown).

Regarding their cognitive abilities, adult SHAM and WIN-exposed animals of both sexes displayed identical discrimination index and exploration time in both the temporal order memory task (*Figure 2E–F*) and the novel object recognition test (*Figure 2G–H*).

Thus, while social interaction was specifically impaired in PCE males, locomotion, anxiety and cognition were spared in both sexes. These data reveal discrete and sex-specific behavioral consequences of PCE at adulthood.

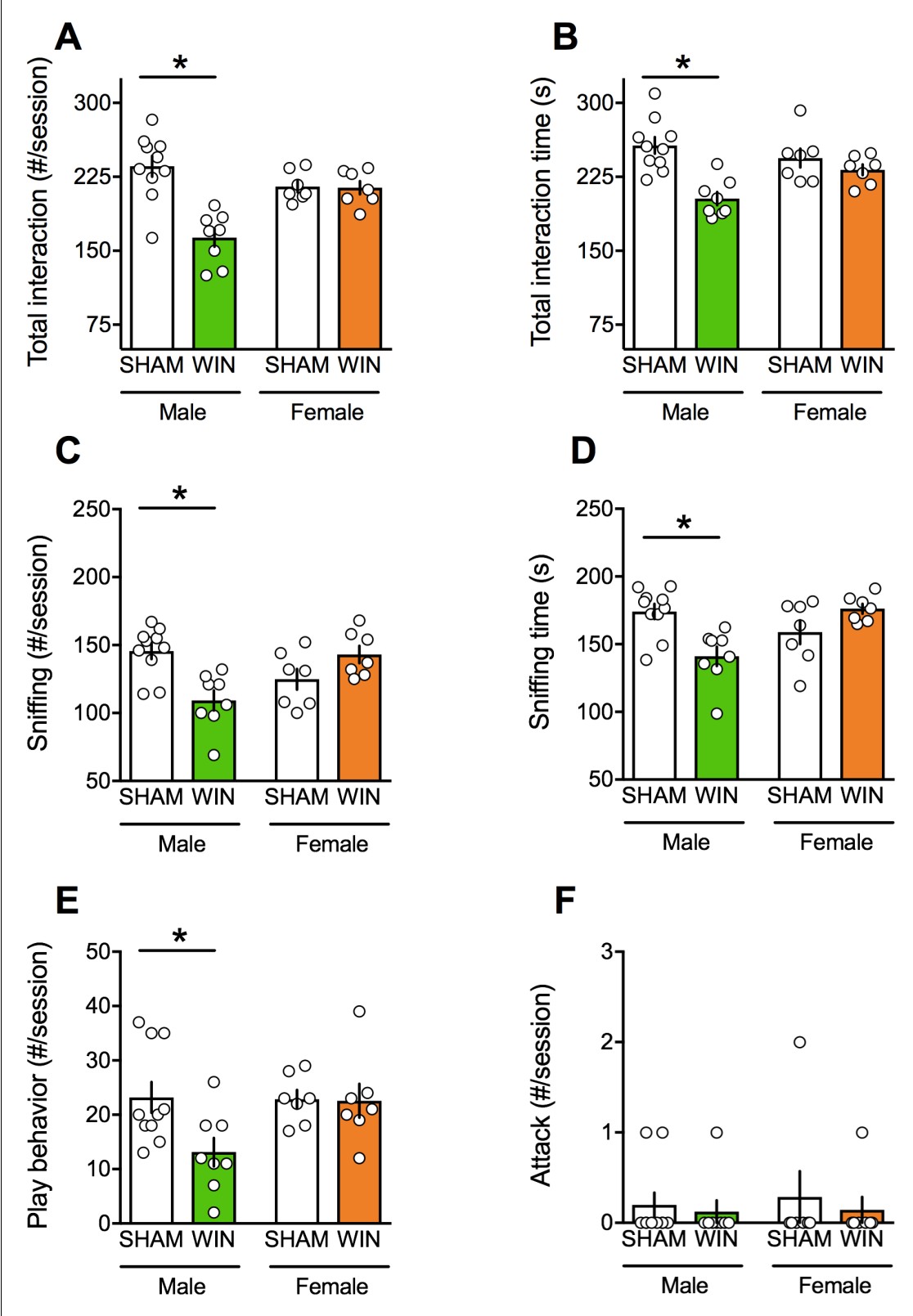

**Figure 1.** Prenatal cannabinoid exposure induces sex-specific social deficits at adulthood. (A–B) Adult male progeny from dams exposed during gestation to WIN had less contact (A): $F_{(sex \times treat)1,28}$=15.54, p<0.05, two-way ANOVA) and spent less time interacting with their congeners than SHAM-animals (B): $F_{(sex \times treat)1,28}$=7.09, p<0.05, two-way ANOVA). In contrast, the social behavior of female littermates was normal. During the social session, the number (C): $F_{(sex \times treat)1,28}$=16.30, p<0.05, two-way ANOVA) and the time (D): $F_{(sex \times treat)1,28}$=14.87, p<0.05, two-way ANOVA) of sniffing events and

*Figure 1 continued on next page*

*Figure 1 continued*

the frequency of play behavior (**E**): $F_{(sex \times treat)1,28}=3.217$, $p>0.05$, two-way ANOVA) were exclusively reduced in male rats exposed to WIN during gestation. Prenatal cannabinoid exposure did not induce aggressive behavior in adult rats (**F**): $F_{(sex \times treat)1,28}=0.037$, $p>0.05$, two-way ANOVA). Scatter dot plot represents a pair of animals. Error bars indicate SEM. *$p<0.05$. Student–Newman–Keuls test.

DOI: https://doi.org/10.7554/eLife.36234.003

The following figure supplement is available for figure 1:

**Figure supplement 1.** Social isolation before testing did not induce sex-specific social changes in adult animals.

DOI: https://doi.org/10.7554/eLife.36234.004

## Prenatal cannabinoid exposure alters synaptic plasticity specifically in the PFC of adult male

In theory, cannabinoid exposure during in-utero neurodevelopment may perturb synaptic functions in most brain areas. The finding of selective impairments of social interaction in the absence of emotional and cognitive deficits does not favor the hypothesis of widespread synaptic deficits. The prefrontal cortex and nucleus accumbens play prominent roles in social behaviors (*van Kerkhof et al., 2013*) and prefronto-accumbens glutamatergic circuits modulate reward-related behaviors (*Floresco, 2015*; *Mateo et al., 2017*). Furthermore, the endogenous cannabinoid (eCB)-system of the accumbens core is instrumental to social interaction behavior (*Manduca et al., 2016*; *Trezza et al., 2012*). In-vivo cannabinoid exposure desensitizes CB1R and ablates eCB-mediated synaptic plasticity in the accumbens (*Mato et al., 2004*; *Mato et al., 2005*). Thus, we first compared eCB-mediated LTD (eCB-LTD) in our experimental groups. The data indicated that eCB-LTD in the accumbens is not affected by PCE (*Figure 3*).

The extensive repertoire of synaptic plasticity displayed by medial prefrontal synapses is a consistent target of environmental and genetic insults (*Iafrati et al., 2014*; *Kasanetz et al., 2013*; *Labouesse et al., 2017*; *Lafourcade et al., 2011*; *Manduca et al., 2017*; *Thomazeau et al., 2014*). We first established that tetanic stimulation induced a robust eCB-LTD of excitatory synapses onto layer 5 mPFC pyramidal neurons in mPFC slices prepared from adult SHAM male and female rats (*Figure 4A–B–C–D*). In marked contrast to the accumbens, eCB-LTD in the mPFC was absent in WIN-exposed males (*Figure 4A–B*). Strikingly, LTD was normal in PCE female littermates (*Figure 4C–D*). In a second series of experiments a similar sex-specific ablation of eCB-LTD in the progeny of dams exposed to THC, the main psychoactive component of cannabis, was observed (5 mg/kg, s.c. GD5-20; *Figure 5*).

To test if the effects of PCE extend to other forms of plasticity expressed at excitatory mPFC synapses we assessed two critical actors of bidirectional plasticity and reliable prefrontal endophenotypes of neuropsychiatric diseases: $mGlu_{2/3}$-dependent LTD and NMDAR-dependent LTP (*Iafrati et al., 2014*; *Iafrati et al., 2016*; *Kasanetz et al., 2013*; *Thomazeau et al., 2014*). In contrast to the lack of eCB-LTD, $mGlu_{2/3}$-mediated LTD was indistinguishable in SHAM- and WIN-exposed male and female (*Figure 4—figure supplement 1 A-B* ). Similarly, theta-burst stimulation (TBS) induces similar levels of NMDAR-dependent LTP at mPFC pyramidal synapses independent of sex and prenatal treatment (*Figure 4—figure supplement 1 C-D-E-F*). We also confirmed that prenatal THC does not affect prefrontal LTP at adulthood (*Figure 4—figure supplement 2*).

Together the data reveal eCB-LTD in the mPFC as sex-specific target of in-utero cannabinoid exposure.

## Cannabinoid prenatal exposure causes sex-specific modifications in pyramidal neuron properties

Activity-induced modulation of intrinsic excitability is a powerful mean to control cortical circuits (*Bacci et al., 2005*; *Debanne and Poo, 2010*) postulated to contribute to disorders of the CNS. To test the hypothesis that PCE altered prefrontal excitability, intrinsic firing properties of pyramidal neurons in our animal groups were compared. Sexual differences to the effects of prenatal exposure to cannabinoids were visible at this cellular level. Specifically, mPFC neurons in WIN-exposed rats showed impaired membrane reaction profiles in response to a series of somatic current steps (*Figure 6A and E*). Independent of the sex and treatment, the resting potential was similar (*Figure 6B and F*). While the rheobase (*Figure 6C and G*) was decreased in males prenatally

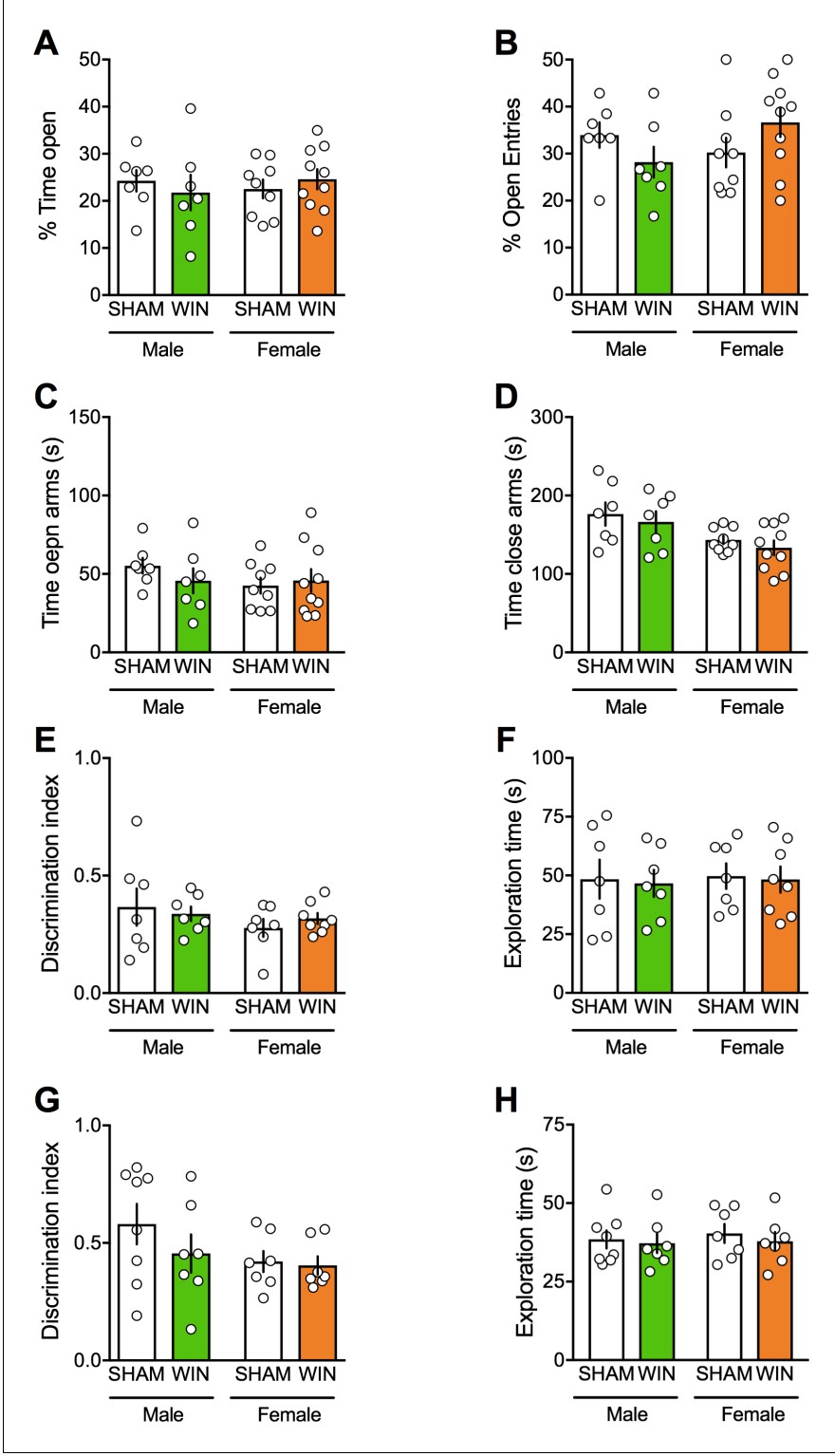

**Figure 2.** Prenatal cannabinoid exposure does not modify adult cognitive and anxious behaviors in both sexes. (A–B) Prenatal cannabinoid exposure did not alter anxiety behavior in adult rats of either sex. No differences between WIN-exposed and SHAM animals were found in the elevated plus maze test as expressed in the percentage of time spent in the open arms (A): $F_{(treat \times sex)1,29}=0.818$, $p>0.05$, two-way ANOVA), the percentage of open arm entries (B): $F_{(treat \times sex)1,29}=3.762$, $p>0.05$, two-way ANOVA), the time spent exploring the open (C): $F_{(treat \times sex)1,29}=0.04$, $p>0.05$, two-way ANOVA) and the closed arm (D): $F_{(treat \times sex)1,29}=0.944$, $p>0.05$, two-way ANOVA). (E–F), Prenatal cannabinoid exposure did not impair cognitive performance. (E–F), In the temporal order memory

*Figure 2 continued on next page*

*Figure 2 continued*

task, no differences were found in the discrimination index (E): $F_{(treat \times sex)1,25}=0.549$, $p>0.05$, two-way ANOVA) and in the total time exploring objects during the test trial (F): $F_{(treat \times sex)1,25}=0.001$, $p>0.05$, two-way ANOVA). (G–H) In the object recognition test, male and female rats exposed to the cannabinoid WIN in-utero showed no differences in their discrimination index (G): $F_{(treat \times sex)1,25}=0.639$, $p>0.05$, two-way ANOVA) or the time exploring objects (H): $F_{(treat \times sex)1,25}=0.04$, $p>0.05$, two-way ANOVA) from the SHAM group, suggesting no deficits in the recognition memory for objects. Scatter dot plot represents each animal. Error bars indicate SEM. *$p<0.05$. Student–Newman–Keuls test.

DOI: https://doi.org/10.7554/eLife.36234.005

exposed to WIN, females were spared. Moreover, this hyper-excitability in exposed-males was accompanied by an increased number of action potentials in response to somatic currents steps (*Figure 6D*). Similar results have been obtained in the rat progeny of dams treated with THC during gestation (*Figure 7*). Thus, PCE leads to an increased excitability of deep layers mPFC pyramidal cells in male progeny only.

## Sex-differences in mRNA expression levels

Motivated by the eCB-LTD deficiency in layer 5 mPFC of the male PCE progeny, we surveyed mRNA levels of key components of the endocannabinoid system in adult PCE rats using real time quantitative PCR (*Figure 8* and *Table 2*). Key differences were decreased levels of mRNA for TRPV1, mGlu5, and DAGLα in the female PCE offspring and decreased levels of mGlu5 mRNA in the male offspring (*Figure 8*). A mild increase was seen in mGlu1 in the PCE males (*Table 2*). No differences were seen in other endocannabinoid system mRNAs after PCE, including CB1, CB2, DAGLβ, NAPE-PLD, FAAH, ABHD6, or CRIP1a (*Table 2*).

## Positive allosteric modulation of mGlu$_5$ corrects synaptic and behavioral deficits associated with prenatal cannabinoid male exposure

The present data reveal a down-regulation of mGlu$_5$ RNA expression levels in the mPFC of PCE males. Previous work from our group underlies the efficiency of mGlu$_5$ positive allosteric modulation in correcting synaptic and behavioral deficits (*Manduca et al., 2017*; *Martin et al., 2017*).

Thus, we tested whether amplification of mGlu$_5$, an effector of eCB-signaling complex in the PFC, could normalize LTD in PCE male. To this aim, we used CDPPB (10 μM), a well-described positive allosteric modulator (PAM) of mGlu$_5$. We found that CDPPB fully restored the ability of excitatory mPFC synapses to express LTD in the prenatally WIN-exposed male group (*Figure 9A–B*). To verify that the curative properties of CDPPB were indeed due to eCB production, we tested the effects of CB1R and TRPV1 antagonists on the CDPPB rescue. Interestingly, preincubation with either SR141716A or capsazepine prevented the CDPPB rescue of synaptic plasticity (*Figure 9C–D*). Thus, the data showed that both receptors are engaged in the presence of the mGlu$_5$ PAM.

Strikingly, we found that systemic CDPPB normalized social cognition in the PCE group but remained without effect in the SHAM group, indicating selectivity of the drug's effects to the disease-state (*Figure 9E–F*). In agreement with the synaptic plasticity experiments, pre-treatment with either SR141716A (*Figure 9G–H*) or capsazepine prevented the ameliorative actions of CDPPB (*Figure 9G–H*). Thus, potentiating mGlu$_5$ signaling normalized synaptic and behavioral deficits induced by prenatal exposure to cannabinoids via CB1R and TRPV1 activation.

## TRPV1, not CB1R, mediate LTD in PCE female

The observation that PCE did not alter CB1R RNA levels in either sex but lowered TRPV1R in the female PCE progeny (*Figure 8*, *Table 2*) drew our attention to the role of these two receptors at PFC synapses. Comparing full dose-response curves for the CB1 agonist CP55,940 in male SHAM vs PCE revealed no differences (*Figure 4—figure supplement 3*). Thus, the lack of LTD in PCE males cannot readily be attributed to altered sensitivity of presynaptic CB1R. We and others have shown that central excitatory synapses can engage TRPV1R and/or CB1R to trigger LTD (*Chávez et al., 2010*; *Grueter et al., 2010*; *Martin et al., 2017*; *Puente et al., 2011*). In PCE females, LTD was fully expressed in the presence of a specific CB1R antagonist (SR141716A 5 μM, *Figure 10A–B–C*) but completely prevented by TRPV1R antagonists (capsazepine 10 μM, *Figure 10A–B–C*). Thus, TRPV1R

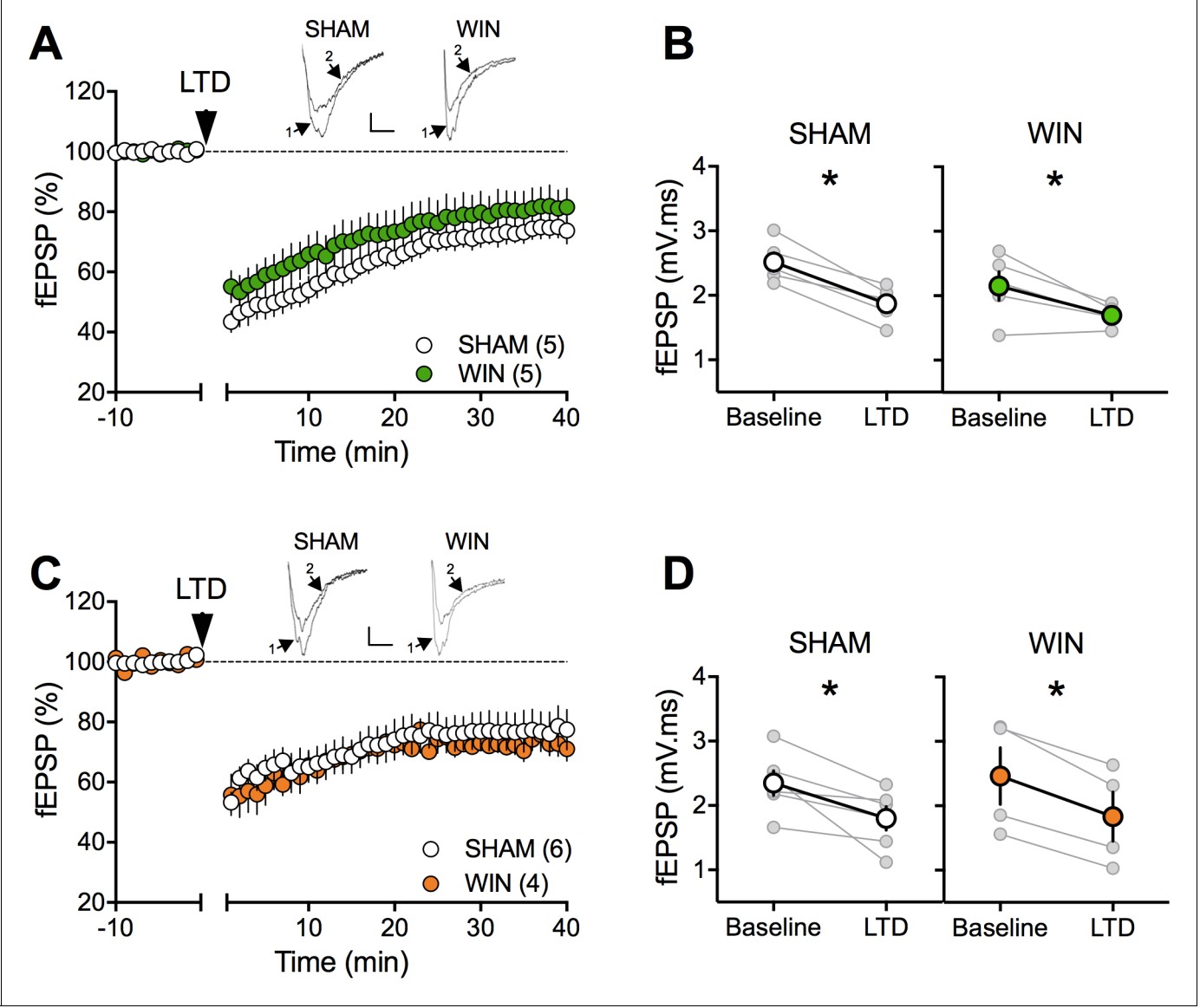

**Figure 3.** Prenatal cannabinoid exposure does not modify LTD in NAc of adult rats. (A) Average time-courses of mean fEPSPs showing that low-frequency stimulation (indicated by arrow) induces LTD at accumbens synapses in both SHAM- (white circles, n = 5) and WIN- (green circles, n = 5) exposed male rats. Above: example traces, baseline (1) and 40 min poststimulation (2). (B) Individual experiments (grey) and group average (tan), before (baseline) and after (40 min) LTD induction showing that PCE does not affect eCB-LTD in male accumbens. In SHAM male rats: 2.518 ± 0.145 mV.ms before LTD versus 1.872 ± 0.124 mV.ms after LTD induction (n = 5, p<0.05, paired t-test). In WIN male: 2.152 ± 0.224 mV.ms before LTD versus 1.696 ± 0.073 mV.ms after LTD induction (n = 5, p<0.05, paired t-test). (C) Average time-courses of mean EPSPs showing similar low-frequency LTD in SHAM (n = 6; white circles) and WIN (n = 4; orange circles) in utero-exposed female rats. Above: example traces, baseline (1) and 40 min poststimulation (2). (D) Individual experiments (grey) and group average (white represents SHAM; orange represents WIN), before (baseline) and after (40 min) LTD induction showing that prenatal WIN exposure does not alter LTD in the female exposed group. In SHAM female: 2.350 ± 0.191 mV.ms before LTD versus 1.800 ± 0.182 mV.ms after LTD induction (n = 6, p<0.05, paired t-test). In WIN female: 2.461 ± 0.439 mV.ms before LTD versus 1.833 ± 0.381 mV.ms after LTD induction (n = 4, p<0.05, paired t-test). n = individual rats. Error bars indicate SEM. Scale bar: 10 ms, 0.1 mV.
DOI: https://doi.org/10.7554/eLife.36234.006

rather than CB1R mediates LTD in female PCE. This contribution of TRPV1R to LTD in female PCE could be part of the adaptive response to PCE or a previously unnoticed sexual difference in the mechanism of prefrontal synaptic plasticity. We next investigated the substrate of LTD in naive male and female rats (*Figure 11*). As expected from over a decade of studies in male rodents (*Bender et al., 2006*; *Chevaleyre et al., 2006*; *Gerdeman et al., 2002*; *Iremonger et al., 2011*;

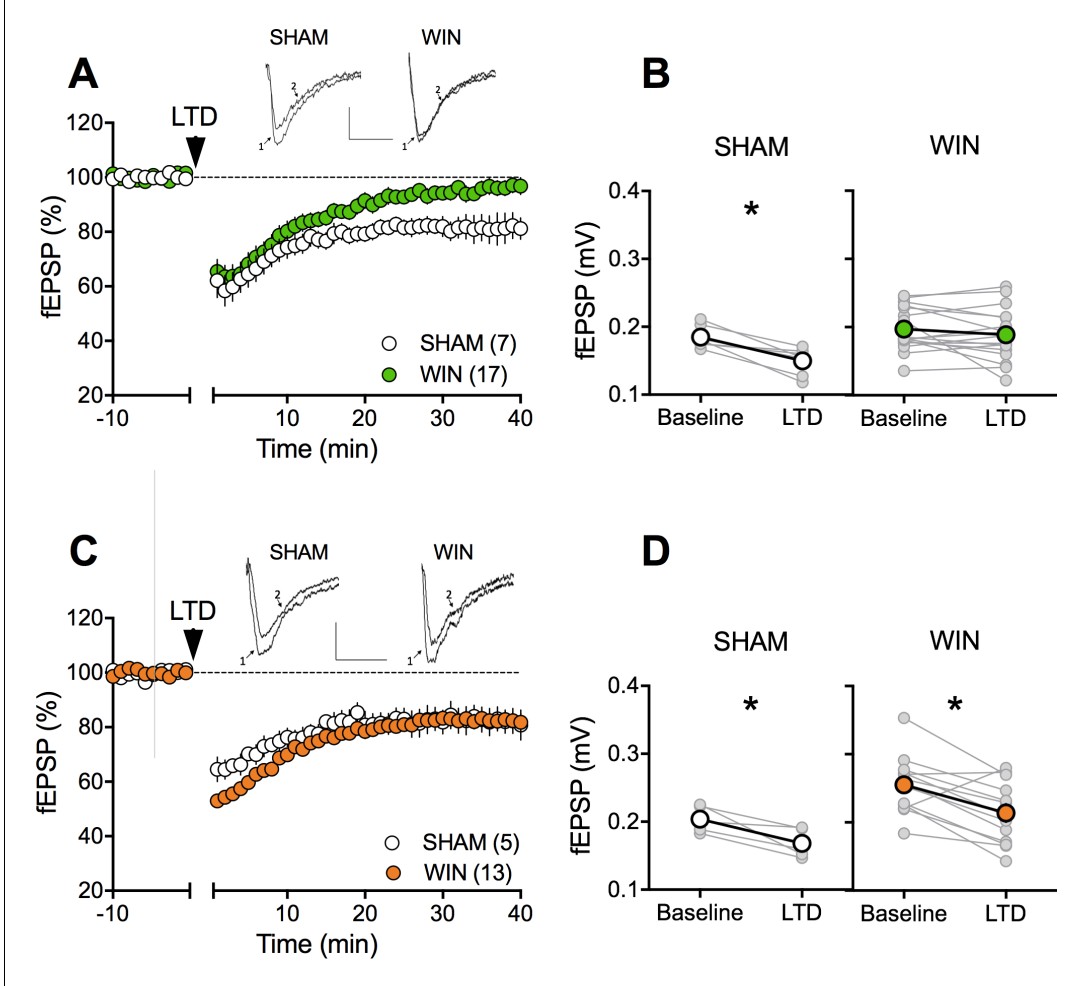

**Figure 4.** Sexual divergent ablation of LTD in the PFC of adult rats prenatally exposed to WIN. (**A**) Average time-courses of mea field EPSPs showing that low-frequency stimulation (10 min at 10 Hz, arrow) induced LTD of evoked fEPSPs recorded in the mPFC in SHAM (n = 7; white circles), but not in WIN (n = 17; green circles) prenatally-exposed male rats. Above: example traces, baseline (1) and 40 min post-stimulation (2). (**B**) Individual experiments (grey) and group average (white represents SHAM; green represents WIN), before (baseline) and after (40 min) LTD induction showing the lack of LTD in the WIN exposed group. In SHAM rats: 0.185 ± 0.006 mV before LTD versus 0.150 ± 0.007 mV after LTD induction (n = 7, p<0.05, paired t-test). In WIN animals: 0.197 ± 0.008 mV before LTD versus 0.189 ± 0.009 mV after LTD induction (n = 17, p>0.05, paired t-test). (**C**) Average time-courses of mean EPSPs showing similar low-frequency LTD in SHAM (n = 5; white circles) and WIN (n = 13; orange circles) in utero-exposed female rats. Above: example traces, baseline (1) and 40 min post-stimulation (2). (**D**) Individual experiments (grey) and group average (white represents SHAM; orange represents WIN), before (baseline) and after (40 min) LTD induction showing that prenatal WIN exposure does not alter LTD in the female exposed group. In SHAM female rats: 0.204 ± 0.009 mV before LTD versus 0.168 ± 0.010 mV after LTD induction (n = 5, p<0.05, paired t-test). In WIN female rats: 0.255 ± 0.012 mV before LTD versus 0.214 ± 0.013 mV after LTD induction (n = 13, p<0.05, paired t-test). *p<0.05. n = individual rats. Error bars indicate SEM. Scale bar: 10 ms, 0.1 mV.

DOI: https://doi.org/10.7554/eLife.36234.007

The following figure supplements are available for figure 4:

**Figure supplement 1.** mGlu$_{2/3}$-LTD and long-term potentiation are spared by prenatal WIN exposure.
DOI: https://doi.org/10.7554/eLife.36234.008
**Figure supplement 2.** Prenatal THC does not alter long-term potentiation.
DOI: https://doi.org/10.7554/eLife.36234.009
**Figure supplement 3.** Prenatal cannabinoid exposure does not modify synaptic CB1R potency or efficiency.
DOI: https://doi.org/10.7554/eLife.36234.010

*Lafourcade et al., 2007*; *Marsicano et al., 2002*; *Nevian and Sakmann, 2006*; *Puente et al., 2011*; *Robbe et al., 2002a*) the CB1R antagonist prevented LTD induction in rat male slices (*Figure 11D,E, F* and *Figure 11— figure supplement 1)*). Unexpectedly (*Rubino et al., 2015*), the same treatment

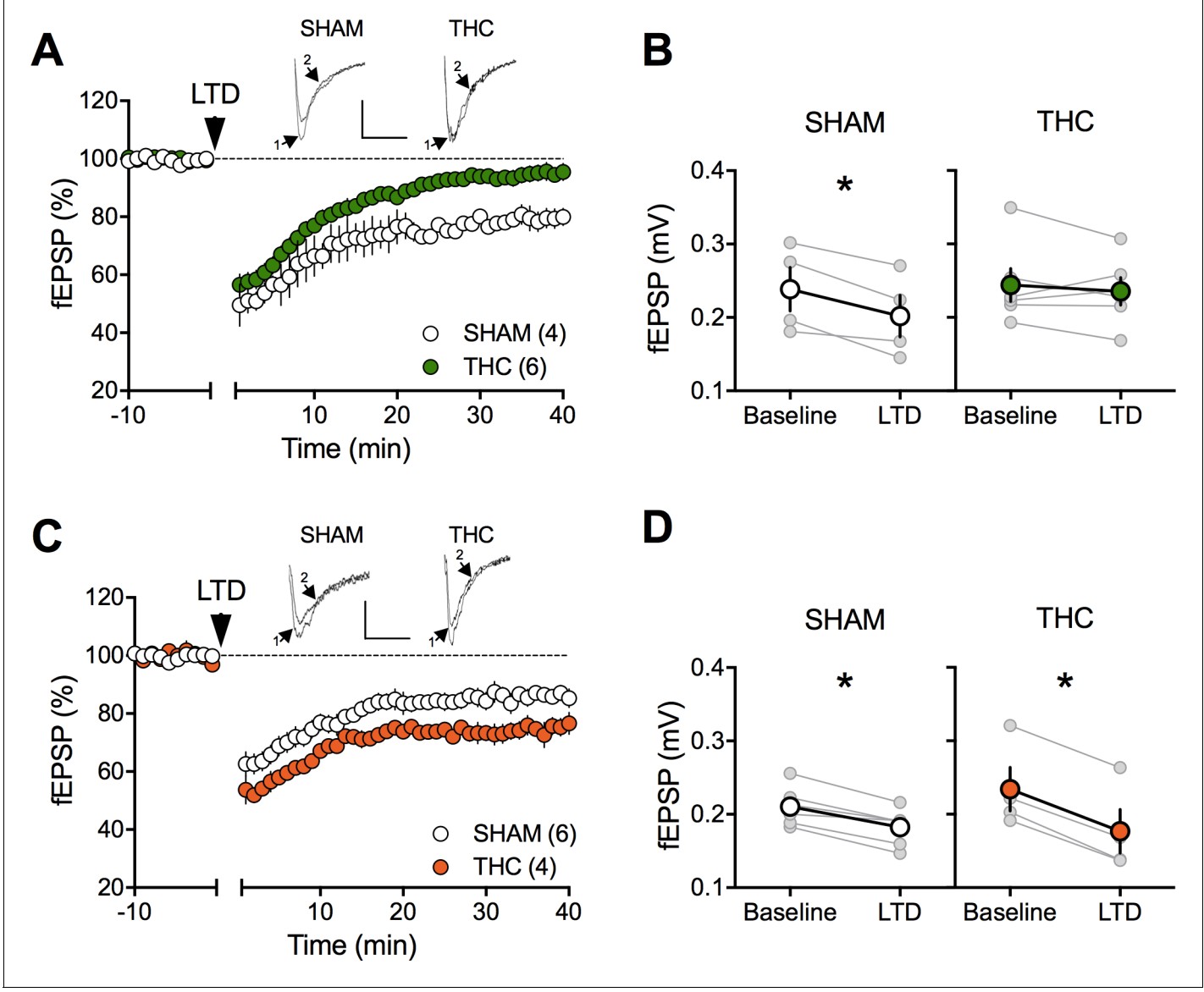

**Figure 5.** Sexual divergent ablation of LTD in the PFC of adult rats prenatally exposed to THC. (**A**) Average time-courses of men field EPSPs showing that low-frequency stimulation (10 min at 10 Hz, arrow) induced LTD of evoked fEPSPs recorded in the mPFC in SHAM (n = 4; white circles), but not in THC (n = 6; green circles) prenatally-exposed male rats. Above: example traces, baseline (1) and 40-min poststimulation (2). (**B**) Individual experiments (grey) and group average (white represents SHAM; green represents THC), before (baseline) and after (40 min) LTD induction showing the lack of LTD in the THC exposed group. In SHAM rats: 0.239 ± 0.030 mV before LTD versus 0.202 ± 0.028 mV after LTD induction (n = 4, p<0.05, paired t-test). In THC animals: 0.244 ± 0.023 mV before LTD versus 0.236 ± 0.019 mV after LTD induction (n = 6, p>0.05, paired t-test). (**C**) Average time-courses of mean EPSPs showing similar low-frequency LTD in SHAM (n = 6; white circles) and THC (n = 4; orange circles) in utero-exposed female rats. Above: example traces, baseline (1) and 40 min poststimulation (2). (**D**) Individual experiments (grey) and group average (white represents SHAM; orange represents THC), before (baseline) and after (40 min) LTD induction showing that prenatal THC exposure does not alter LTD in the female exposed group. In SHAM female rats: 0.211 ± 0.011 mV before LTD versus 0.183 ± 0.010 mV after LTD induction (n = 6, p<0.05, paired t-test). In THC female rats: 0.234 ± 0.030 mV before LTD versus 0.177 ± 0.030 mV after LTD induction (n = 4, p<0.05, paired t-test). *p<0.05. n = individual rats. Error bars indicate SEM. Scale bar: 10 ms, 0.1 mV.

DOI: https://doi.org/10.7554/eLife.36234.011

was without effect in female slices (*Figure 11A–B–C*). In contrast, antagonism of TRPV1R was without effect in males (*Figure 11A–C*) but prevented the induction of eCB-LTD in females (*Figure 11D–E–F*).

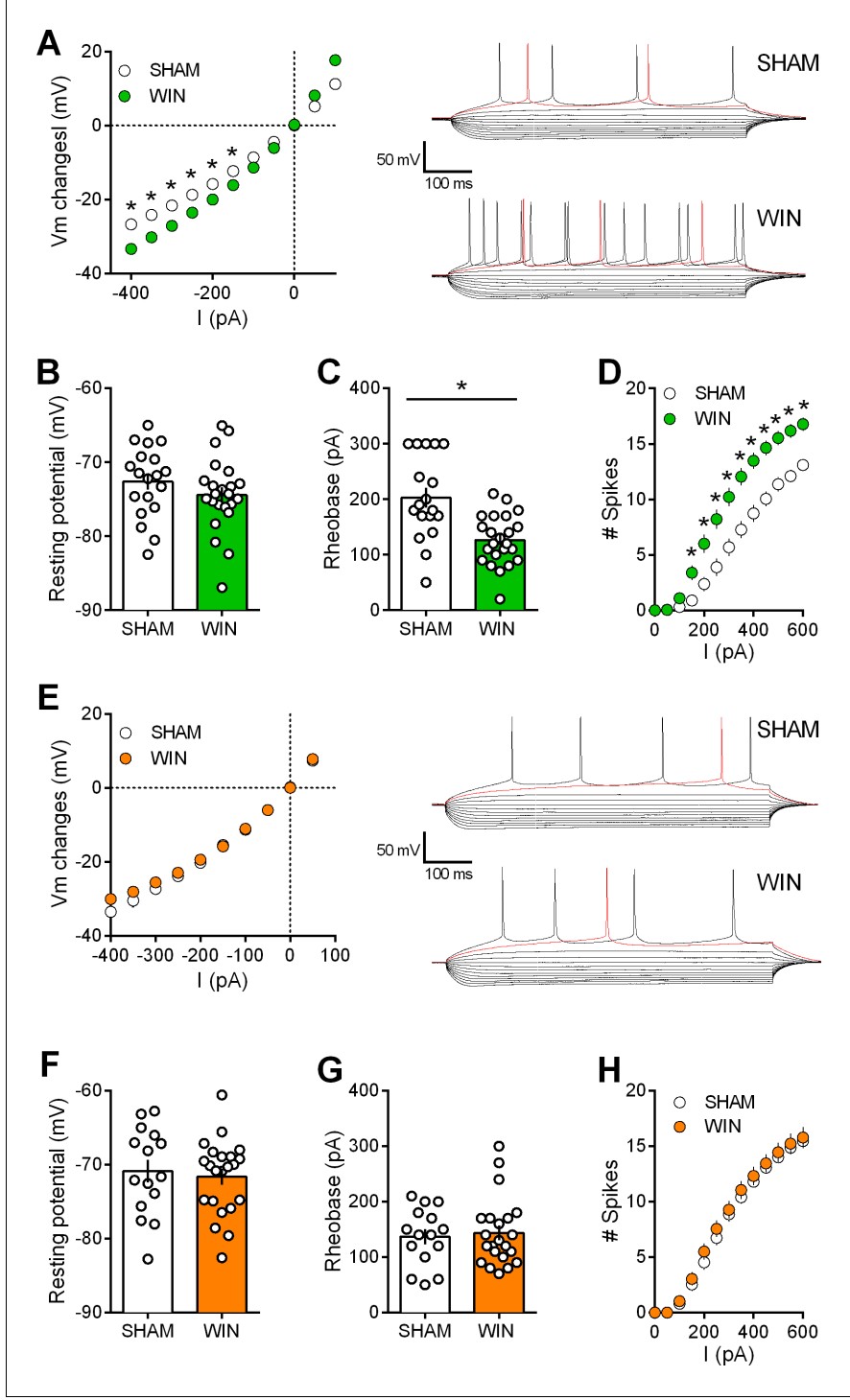

**Figure 6.** Prenatal WIN exposure induces sex-specific alteration of intrinsic properties of layer five prefrontal pyramidal neurons in adult rats. (**A**) Left, Current-voltage (I–V) curves recorded in SHAM (n = 19 cells/10 rats, white circles) and WIN-exposed rats (n = 23 cells/12 rats, green circles) showing altered membrane potentials in male from the in utero cannabinoid group (F$_{(10, 400)}$=10.07, p<0.05, two-way ANOVA). Right, typical membrane response to somatic current steps from SHAM (up) and WIN- (down) exposed male. (**B**) The resting membrane potential was similar in both SHAM and WIN male groups (p>0.05, Mann-Whitney U test). (**C**) The rheobase was reduced in WIN-exposed male rats compared to the SHAM male (p<0.05, Mann-Whitney U test). (**D**) The number of evoked action potentials in response to increasing depolarizing current steps was higher in WIN in-utero treated male compared to SHAM animals (F$_{(18, 720)}$=6.652, p<0.05, two-way ANOVA). (**E**) Left, In contrast, I-V curves were similar

*Figure 6 continued on next page*

*Figure 6 continued*

in SHAM (n = 15 cells/9 rats, white circles) and WIN-exposed female (n = 22 cells/12 rats, orange circles; $F_{(9, 315)}$=1.091, p>0.05, two-way ANOVA). Right, typical membrane response to somatic current steps from SHAM (up) and WIN- (down) exposed females. (**F,G**) The resting membrane potential and the rheobase were not impacted in females (**F,G**) p>0.05, Mann-Whitney U test). (**H**) The number of action potentials was not altered in females after an in utero treatment ($F_{(12, 455)}$=0.079, p>0.05, two-way ANOVA). *p<0.05. Error bars represent SEM.

DOI: https://doi.org/10.7554/eLife.36234.012

We confirmed and extended these findings with two other antagonists: the CB1R selective neutral antagonist NESS0327 (1 µM, Cayman Chemical, *Figure 11—figure supplement 1A-B*) but not the selective TRPV1R antagonist AMG9810 (3 µM, Tocris, *Figure 11—figure supplement 1A-B*) prevented LTD induction in naive male rat slices. Conversely, AMG9810 but not NESS0327 prevented the induction of LTD in naive female adult animals (*Figure 11—figure supplement 1 C-D* ).

## Enhancing anandamide levels corrects synaptic and behavioral deficits associated with prenatal cannabinoid male exposure

We thought of an additional strategy to reverse PCE-induced deficits based on our finding that TRPV1R mediates LTD in PCE females in spite of a down-regulation of TRPV1R RNA expression levels. Anandamide is an endogenous ligand of TRPV1 (*Di Marzo et al., 2002*) and we hypothesized that inhibition of anandamide's main degrading enzyme, fatty acid amid hydrolase (FAAH), would increase anandamide levels to the threshold for LTD induction in the PCE males (*Rubino et al., 2015*). Indeed, incubation of mPFC slices prepared from PCE male rats with the FAAH inhibitor URB597 effectively restored eCB-LTD (*Figure 12A–B*). Interestingly, we found that pre-incubation with the specific TRPV1 antagonist capsazepine did not prevent the curative effect of URB597. Rather, the CB1R antagonist SR141716 blocked the URB597 rescue (*Figure 12C–D*). Previous studies have shown that endocannabinoids, including anandamide, mediate social behavior in rodents (*Trezza et al., 2012*). We directly tested the hypothesis that elevating circulating levels of a main eCB, anandamide, could normalize social deficits. We found that in vivo treatment with the FAAH inhibitor (URB597, 0.1 mg/kg) normalized social deficits in adult male PCE rats (*Figure 12E–F*). In line with our mPFC slice results, SR141716A but not capsazepine prevented the curative actions of URB597 (*Figure 12G–H*). Thus, blocking anandamide degradation normalized social deficits via CB1R.

## Discussion

We discovered that fetal exposure to cannabinoids causes sex-specific alterations of behavioral and synaptic functions in the adult progeny. Specifically, we found that prenatal cannabinoid exposure (PCE) reduces social interactions in males but not females, while other behaviors were spared. In parallel, PCE specifically altered neuronal excitability and synaptic plasticity in the prefrontal cortex of male rats. We also identify a pharmacological strategy to normalize synaptic functions and socialization in adult prenatally exposed males.

PFC malfunctions are a common denominator in several neuropsychiatric diseases (*Goto et al., 2010*; *Iafrati et al., 2014*; *Kasanetz et al., 2013*; *Labouesse et al., 2017*; *Lafourcade et al., 2011*; *Manduca et al., 2017*; Martin et al.). Our data indicate that in the PCE group, plasticity deficits were circumscribed to eCB-LTD as two other types of synaptic plasticity, which act as biomarkers for prefrontal deficits in other model of neuropsychiatric diseases, were spared. Namely neither NMDAR-mediated LTP (*Iafrati et al., 2014*; *Manduca et al., 2017*; *Martin and Manzoni, 2014*; *Thomazeau et al., 2017*) nor mGluR2/3-LTD (*Kasanetz et al., 2013*) were affected in PCE males. Furthermore, while excitatory synapses of the nucleus accumbens display reduced eCB-mediated plasticity in models of depression (*Bosch-Bouju et al., 2016*), autism (*Jung et al., 2012*), adult cannabis exposure (*Mato et al., 2004*; *Mato et al., 2005*) or nutritional deficits (*Manduca et al., 2017*), accumbal eCB-LTD was normal in both PCE males and females.

Our behavioral exploration reveals that social interactions were reduced in males after PCE while female social behavior and other behavioral domains were spared in both sexes. Our results greatly extend that of Vargish and collaborators, who observed reduced social interaction in PCE mice (*Vargish et al., 2017*). In the latter study, female progeny was unfortunately not tested. The current

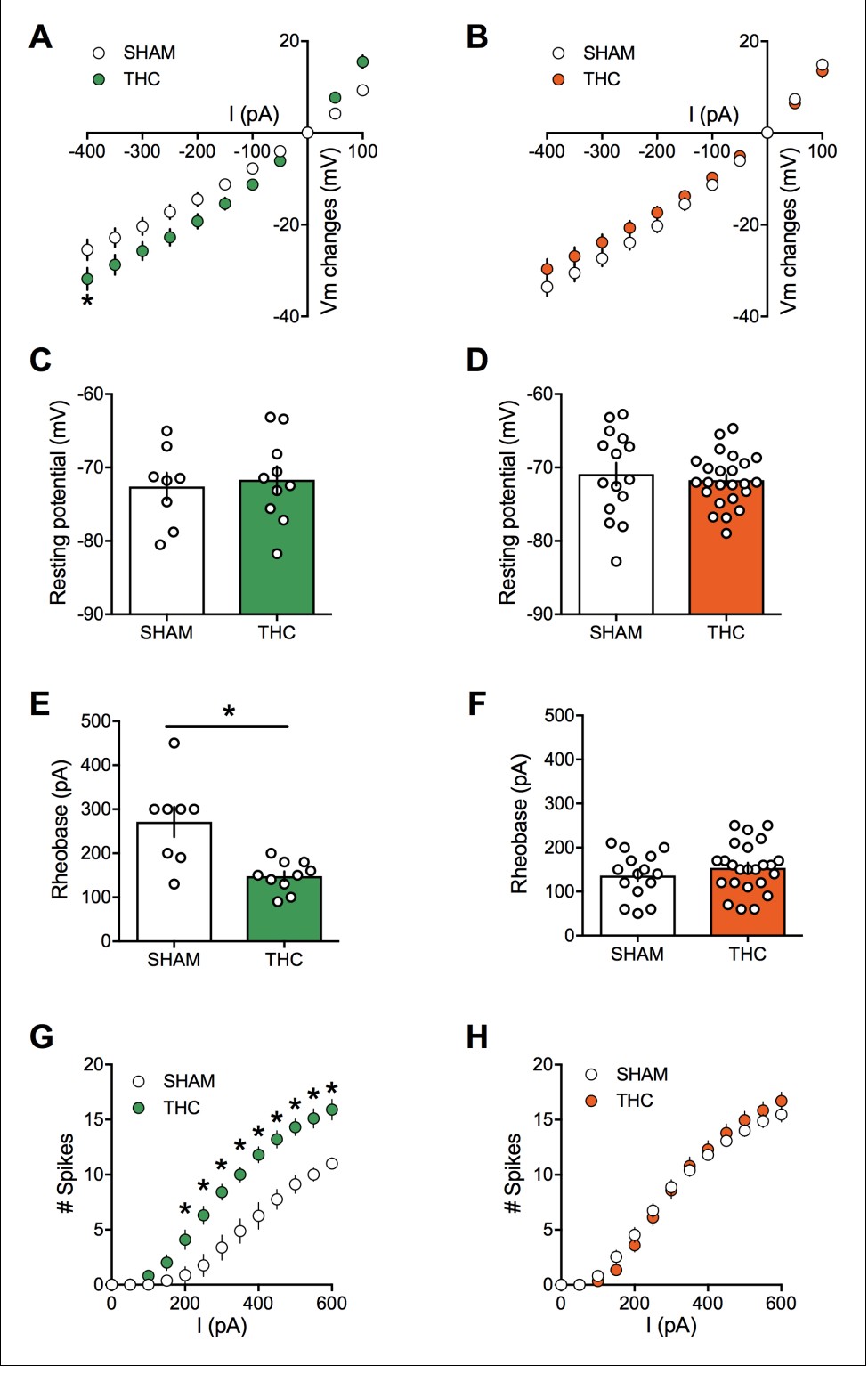

**Figure 7.** Prenatal THC exposure induces sex-specific alteration of intrinsic properties of layer five prefrontal pyramidal neurons in adult rats. (**A**) Current-voltage (I–V) curves recorded in SHAM (n = 8 cells/5 rats, white circles) and THC-exposed rats (n = 10 cells/5 rats, green circles) showing altered membrane potentials in male after an in utero cannabinoid exposure ($F_{(26, 416)}$=1.987, p<0.05, two-way ANOVA). (**B**): In contrast, I-V curves were similar in SHAM group (n = 15 cells/7 rats, white circles) and THC-exposed group (n = 24 cells/7 rats, orange circles; $F_{(26, 962)}$=0.534, p>0.05, two-way ANOVA). (**C,D**): The resting membrane potential was similar in both male (**C**) and

*Figure 7 continued on next page*

*Figure 7 continued*

female (**D**) rats after an in utero exposure to THC (p>0.05, Mann-Whitney U test). (**E,F**): The rheobase was reduced in THC-exposed male rats compared to the SHAM male (**E**); p<0.05, Mann-Whitney U test) whereas it was not altered in female (**F**); p>0.05, Mann-Whitney U test). (**G**): The number of evoked action potentials in response to several depolarizing current steps was higher in THC in utero treated male compared to SHAM animals ($F_{(18, 288)}$=3.764, p<0.05, two-way ANOVA). (**H**): The number of action potential was also altered in female after an in utero treatment with THC ($F_{(18, 666)}$=1.320, p<0.05, two-way ANOVA). Error bars represent SEM.

DOI: https://doi.org/10.7554/eLife.36234.013

results add to previous work showing covariation between defects in PFC synaptic plasticity and social interaction behaviors (*Manduca et al., 2017*). Together, these data suggest a certain level of regional PFC specificity to the effects of PCE. The fact that the PFC develops late in to postnatal life (i.e. late adolescence/early adulthood) (*Arain et al., 2013*; *Fuster, 1991*; *Kolb et al., 2012*) may explain this sensitivity.

In the hippocampus, PCE impaired eCB-dependent short-term depression at GABAergic synapses (*Vargish et al., 2017*) and NMDAR-dependent LTD (*Mereu et al., 2003*; *Tortoriello et al., 2014*). In contrast with the latter hippocampal study, we found no evidence that PCE modified release properties of PFC synapses, reinforcing the idea that region-specific mechanisms engage to alter PFC synapses in response to PCE. Our qRT-PCR data showed similar levels of CB1R mRNA in the PFC of male and female progeny exposed to cannabinoids during fetal life, in line with previous observations in the hippocampus and neocortex (*Tortoriello et al., 2014*).

While our work clearly shows previously undisclosed sexual divergence in the consequences of fetal cannabinoids, they also reveal sex differences in the main elements involved in eCB signaling: CB1R was similar in both sexes while lower mRNA levels of the 2-arachidonoylglycerol (2-AG) synthesizing (DAGLα) enzyme were found in naive females compared to naive males. Previous work highlighted sexual differences at the molecular level. CB1R affinity is higher in males while receptor density is similar in both sexes in the limbic forebrain; however, in the mesencephalon both parameters are higher in males (*Rodríguez de Fonseca et al., 1994*). In the PFC, CB1R density is lower in males than in females (*Castelli et al., 2014*). At the structural level, the medial PFC of rats is also sexually different: female prelimbic cortex pyramidal neurons have smaller/less complex apical dendritic arbors than in males (*Garrett and Wellman, 2009*; *Kolb and Stewart, 1991*). Thus, it is likely that both nanoscale differences in the relative position of the molecular elements of the eCB system and dissimilarities of mRNA/protein expression underlie these sex differences. At the behavioral level, sexual differences in fear-conditioning have also been identified (*Gruene et al., 2015*; *Gupta et al., 2001*; *Pryce et al., 1999*). Sexual differences are unlikely exclusive to the eCB-LTD/PFC/fear conditioning. For instance, active avoidance (*Dalla and Shors, 2009*), spatial memory (*Dalla and Shors, 2009*; *Qi et al., 2016*) and hippocampal LTP (*Gupta et al., 2001*; *Maren, 1995*; *Qi et al., 2016*) display sex differences. The data provide new impetus for the urgent need to investigate the functional and behavioral substrates of neuropsychiatric diseases in both sexes.

Reduced bioavailability of endocannabinoids during LTD induction likely underlies the lack of synaptic plasticity found in PCE males. In support of this idea, basal levels of anandamide are reduced in limbic areas of prenatally WIN-exposed rats (*Castelli et al., 2007*). We hypothesized that enhancing anandamide levels may lower the LTD-induction threshold and indeed found that FAAH inhibition with URB597 restored eCB-LTD in PCE males. Furthermore, based on the well-known, prominent role of mGlu$_5$ in synaptic eCB-signaling (*Araque et al., 2017*), we found that positive allosteric modulation of mGlu$_5$ with CDPPB (*Manduca et al., 2017*; *Martin et al., 2016a*) restored LTD in PCE males. Strikingly, this finding translated well from the ex vivo slice preparations to the behavioral level, and in vivo treatments with either the FAAH inhibitor or the mGlu$_5$ PAM normalized social interactions in PCE exposed adult males.

In conclusion, these results provide compelling evidences for sexual divergence in the long-term functional and behavioral consequences of PCE and introduce strategies for reversing its detrimental effects.

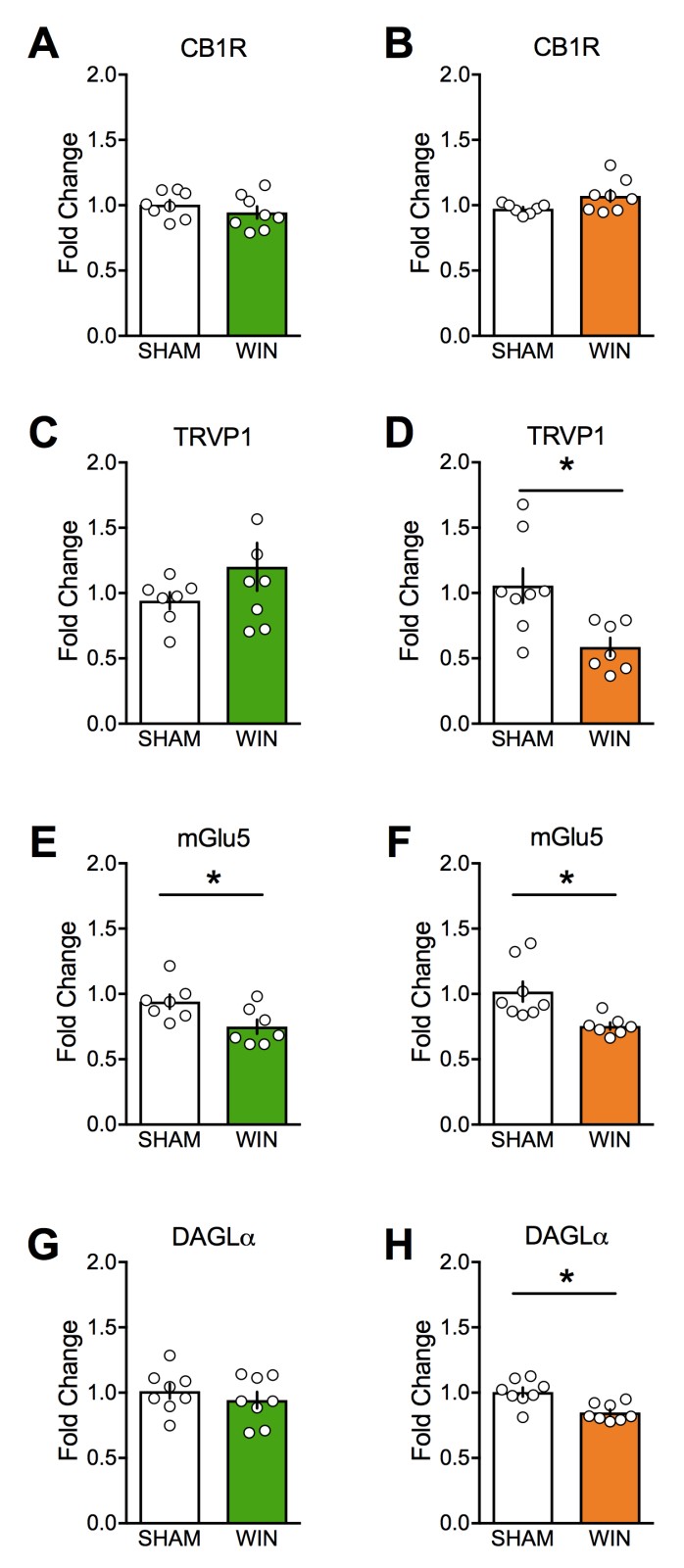

**Figure 8.** Prenatal cannabinoid treatment affects expression of endocannabinoid system components in a sex-dependent fashion. CB1, TRPV1, mGlu5, and DAGLα mRNA levels in medial prefrontal cortex at PND90 were determined by real-time PCR in male (**A, C, E, and G**) and female (**B, D, F, and H**) rats using Taqman probes and a QuantStudio7 thermocycler. Treatment of dams with WIN from GD5 to GD20 decreased TRPV1, mGlu5, and DAGLα mRNA in female offspring. However, the same treatment decreased only mGlu5 in male offspring. *p<0.05. Error bars represent SEM.

*Figure 8 continued on next page*

*Figure 8 continued*

DOI: https://doi.org/10.7554/eLife.36234.014

## Materials and methods

### Animals

The experiments were performed following the ARRIVE (Animals in Research: Reporting In Vivo Experiments) guidelines (*Kilkenny et al., 2010*). Wistar female rats (Charles River, France) weighing 250 – 280 g were housed at constant room temperature (20 ± 1°C) and humidity (60%), and exposed to a light cycle of 12 hr/day (08:00 a.m. to 08:00 p.m.), with food and water available ad libitum. For mating, pairs of females were placed with single male rat in the late afternoon. Vaginal smears were

**Table 2.** Statistical significance of gene expression changes in adult rat medial prefrontal cortex after in utero WIN treatment.

The levels of mRNA's of interest were determined from medical prefrontal cortex punches using qPCR (See Materials and methods for details). Statistical significance was determined using the unpaired Student's t-test after outliers were detected and removed from the dataset using Grubbs' test. p-Values of less than 0.05 are identified by bold text in the table. Relative expression levels for CB1, TRPV1, mGluR5 and DAGL-alpha are shown in *Figure 10*.

| | **MALE** | | | | Female | | |
|---|---|---|---|---|---|---|---|
| **Targets** | **Treatment** | **N** | **p-Value (Unpaired t-test)** | | **Treatment** | **N** | **p-Value (Unpaired t-test)** |
| CB1 | SHAM | 8 | 0.3279 | | SHAM | 7 | 0.0643 |
| | WIN | 8 | | | WIN | 8 | |
| CB2 | SHAM | 7 | 0.9379 | | SHAM | 6 | 0.1427 |
| | WIN | 7 | | | WIN | 7 | |
| TRPV1 | SHAM | 7 | 0.2281 | | SHAM | 8 | **0.0098** |
| | WIN | 8 | | | WIN | 7 | |
| mGlu5 | SHAM | 7 | **0.0273** | | SHAM | 8 | **0.0092** |
| | WIN | 7 | | | WIN | 7 | |
| mGlu1 | SHAM | 8 | **0.0400** | | SHAM | 7 | 0.3910 |
| | WIN | 8 | | | WIN | 8 | |
| DAGL alpha | SHAM | 8 | 0.4370 | | SHAM | 8 | **0.0024** |
| | WIN | 8 | | | WIN | 8 | |
| DAGL beta | SHAM | 7 | 0.4218 | | SHAM | 6 | 0.3783 |
| | WIN | 7 | | | WIN | 7 | |
| NAPE-PLD | SHAM | 7 | 0.4258 | | SHAM | 7 | 0.7518 |
| | WIN | 7 | | | WIN | 7 | |
| MAGL | SHAM | 8 | 0.5097 | | SHAM | 8 | 0.2147 |
| | WIN | 8 | | | WIN | 8 | |
| FAAH | SHAM | 8 | 0.5505 | | SHAM | 8 | 0.4096 |
| | WIN | 8 | | | WIN | 7 | |
| ABHD6 | SHAM | 7 | 0.2437 | | SHAM | 6 | 0.6014 |
| | WIN | 7 | | | WIN | 7 | |
| CRIP1a | SHAM | 7 | 0.6414 | | SHAM | 7 | 0.1887 |
| | WIN | 7 | | | WIN | 7 | |

DOI: https://doi.org/10.7554/eLife.36234.015

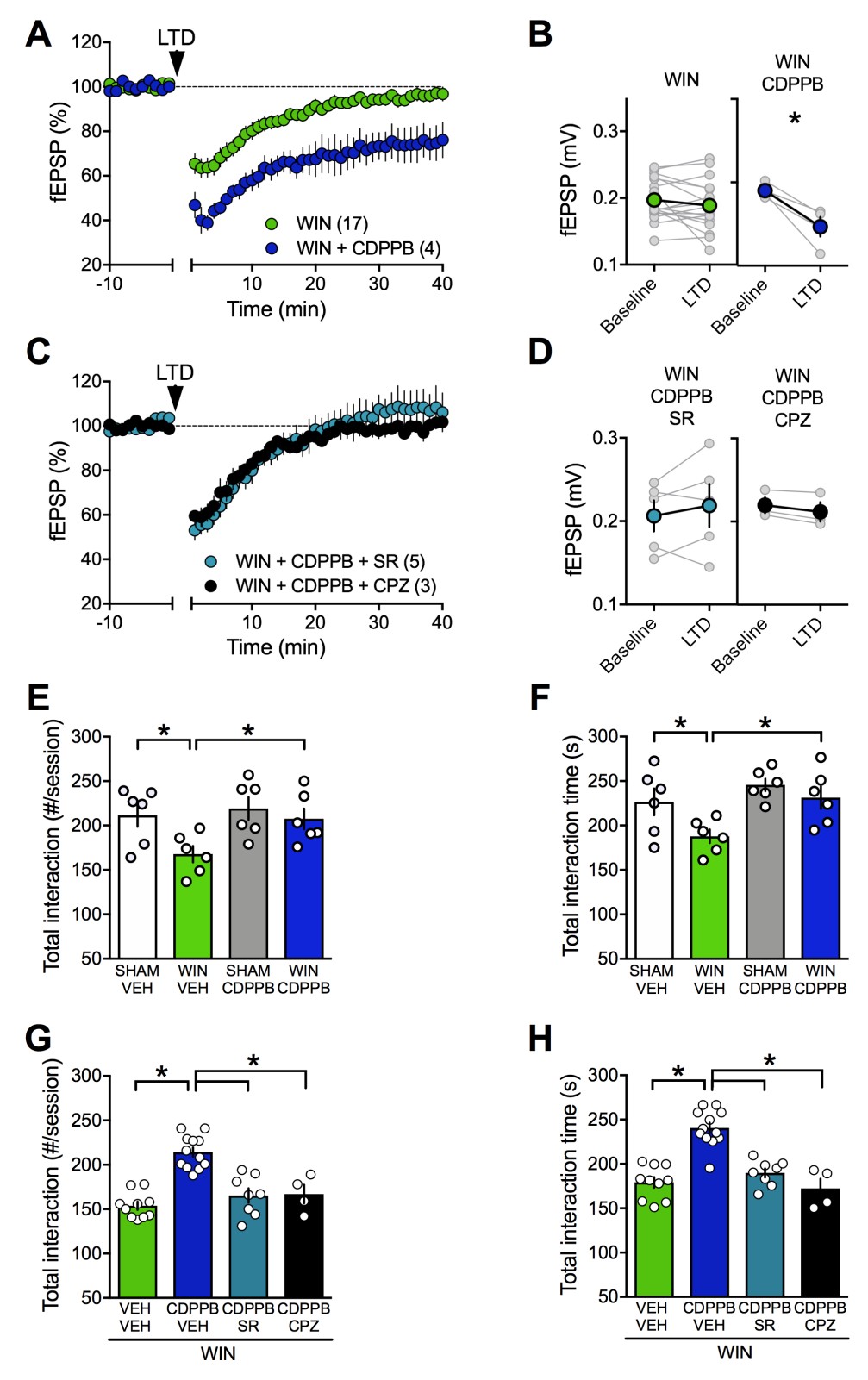

**Figure 9.** Positive allosteric modulation of mGlu$_5$ restores LTD and normalizes social interaction in male rats prenatally exposed to WIN. (**A**) Average time-courses of mean field EPSPs showing that, in WIN-exposed males (WIN, n = 17, green circle), LTD can be restored with the mGlu$_5$positive allosteric modulator CDPPB (10 μM, n = 4, blue navy circle). (**B**) Peak amplitude measurements before (baseline) and after (LTD) stimulation protocol from individual experiments in absence or presence of CDPPB. After 45 min CDPPB preincubation: 0.190 ± 0.004 mV before and 0.147 ± 0.012 after LTD

*Figure 9 continued on next page*

*Figure 9 continued*

induction (n = 4, p<0.05, paired t-test). (C) Time course of fEPSP during 10 Hz stimulus evoked LTD (arrow, 10 min) in presence of CB1R antagonist SR141716 (SR, 4 μM, n = 5, blue circles) or TRPV1 antagonist Capsazepine (CPZ, 10 μM, n = 3, black circles) in WIN-exposed males. (D) Individual averaged fEPSP amplitude experiments before (baseline) and 40 min after 10 Hz LTD stimulation (LTD). CB1R were necessary for the CDPPB effect on LTD: SR141716A preincubation blocked LTD induction in the presence of CDPPB: 0.206 ± 0.019 mV before and 0.217 ± 0.026 (blue circle, n = 5, p>0.05, paired t-test). Similarly, the TRPV1 antagonist inhibits the restorative effects of CDPPB: 0.220 ± 0.009 mV before LTD and 0.212 ± 0.012 mV after LTD induction (black circle, n = 3, p<0.05, paired t-test). (E–F) Systemic administration of CDPPB (0.75 mg/Kg, i.p.) normalized the altered social behavior in male rats prenatally exposed to WIN (number of contacts: E: $F_{(WIN\ in\ utero\ \times\ CDPPB)1,20}$=1.867, p=0.187, two-way ANOVA; time interaction: F: $F_{(WIN\ in\ utero\ \times\ CDPPB)1,20}$=1.607, p=0.219, two-way ANOVA). (G-H) Pre-treatment with SR141716A (1 mg/kg, i.p.) prevents the ameliorative actions of CDPPB (0.75 mg/Kg, i.p.) on social interaction. Similarly, the TRPV1 antagonist blocks the restorative effects of CDPPB on social interaction in adult rats prenatally exposed to WIN (number of contacts: G: $F_{3,30}$=22.36, p<0.05, one-way ANOVA; time interaction: H: $F_{3,30}$=25.95, p<0.05, one-way ANOVA). Scatter dot plot represents a pair of animals. Error bars indicate SEM. *p<0.05. Student–Newman–Keuls test.
DOI: https://doi.org/10.7554/eLife.36234.016

taken the following morning at 09:00 a.m. The day on which sperm was present was designated as the day 0 of gestation (GD 0).

Sub-cutaneous (s.c.) injections were performed daily with the synthetic cannabinoid WIN55,212 – 2 (WIN, 0.5 mg/kg), the phytocannabinoid Delta9-Tetrahydrocannabinol, THC (5 mg/kg) or their respective vehicles from GD 5 to GD 20. WIN was suspended in 5% DMSO, 5% cremophor and saline, at a volume of 1 ml/kg. Control dams (SHAM group) received a similar volume injection of vehicle solution. THC was suspended in 5% Ethanol, 5% cremophor and saline, and injected s.c. at a volume of 1 ml/kg. Control dams (SHAM group) received a similar volume injection of vehicle solution. Newborn litters found up to 05:00 p.m. were considered to be born on that day (postnatal day (PND) 0). On PND 21, the pups were weaned and housed in groups of three. The experiments were carried out on the male and female offspring at adulthood (PNDs 90 – 130). One pup per litter from different litters per treatment group was used in the behavioral experiments. Sample size (n and scatter dot plot) is indicated in the figure legends.

Animals were treated in compliance with the European Communities Council Directive (86/609/EEC) and the United States National Institutes of Health Guide for the care and use of laboratory animals.

Body weights of the dams were taken daily throughout pregnancy and the length of pregnancy was determined. Litter size, weight gain of pups and postnatal vitality were also measured, see *Table 3*.

## Behavior
### All animals were experimentally naïve and used only once
*Social interaction:* Social behavior was assessed as previously described (*Manduca et al., 2015*). The test was performed in a sound attenuated chamber under dim light conditions. The testing arena consisted of a Plexiglas cage measuring 45 × 45 cm, with 2 cm of wood shavings covering the floor. Rats were individually habituated to the test cage for 5 min on each of the 2 days prior to testing. Before testing, animals were socially isolated for 24 hr to enhance their social motivation and thus facilitate the expression of social behaviors during testing.

We verified that the short period of social isolation before testing does not modify social interaction between males and females (*Figure 1—figure supplement 1*). Naive male and female rats not isolated before testing showed similar frequency and time of social interaction to rats isolated 24 hr prior social interaction testing (*Figure 1—figure supplement 1A-B*) . Detailed exploration of the various parameters of social interaction revealed that sniffing (*Figure 1—figure supplement 1C-D*) was unchanged and the play events were reduced only in female animals (*Figure 1—figure supplement 1E*), while the number of attacks remained unchanged (*Figure 1—figure supplement 1F*).

Drug treatments were counterbalanced so that cage mates were allocated to different treatment groups. The animals of each pair were similarly treated, did not differ more than 10 g in body weight and were not cage mates. Behavior was assessed per pair of animals and analyzed by a trained observer who was unaware of treatment condition using the Ethovision XT 13.0 video tracking software (Noldus, Wageningen, The Netherlands).

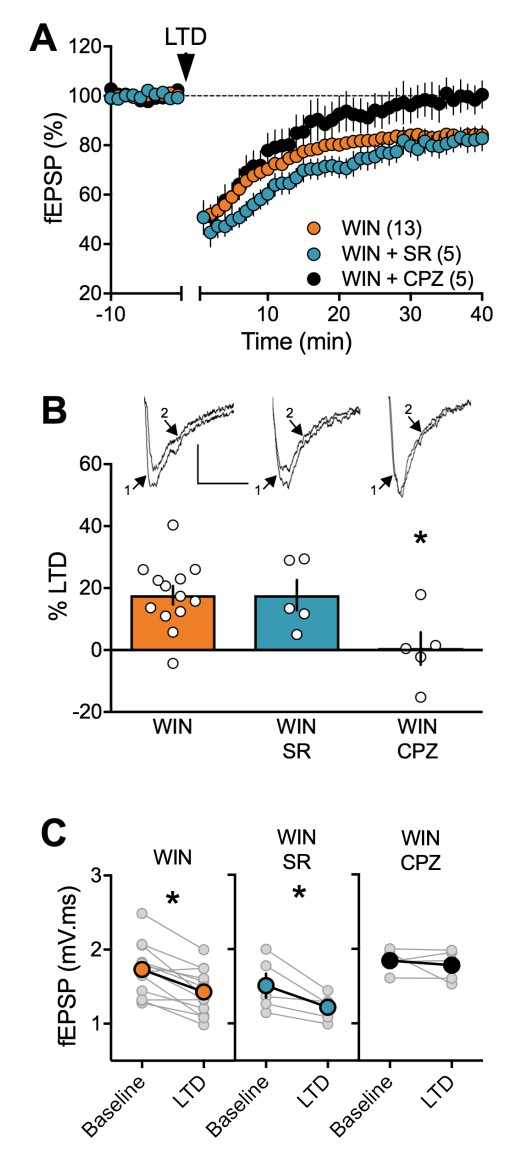

**Figure 10.** In prenatally exposed female, TRPV1 receptors mediate LTD, not CB1R. (**A**) Average time course of mean fEPSPs showing that LTD (arrow) is inhibited by the selective TRPV1 antagonist capsazepine (CPZ, black circles, n = 5) but not the CB1R antagonist SR141617A (SR, blue circles, n = 5) in WIN in utero exposed female rats. (**B**) Summary bar chart of percent LTD calculated from normalized fEPSPs measured 40 min after 10 Hz protocol showing that LTD is blocked by TRPV1 antagonist (black bar) but not by CB1R antagonist (blue bar; p>0.05, one-way ANOVA). Above: example traces, baseline (1) and 40 min poststimulation (2). (**C**) Plot of individual experiments (grey) showing fEPSP areas before (baseline) and 40 min after stimulation protocol (LTD) in absence (orange, n = 13; p<0.05, paired t-test) and presence of SR141716A (SR, blue, n = 5; p<0.05, paired t-test) or Capsazepine (CPZ, black, n = 5; p>0.05, paired t-test) from WIN-exposed female rats. *p<0.05.

*Figure 10 continued on next page*

The test consisted of placing two similarly treated animals into the test cage for 10 min. The total frequency of active social interactions was obtained as the sum of the frequency of the following behavioral elements scored per 10 min (*Manduca et al., 2015*; *Segatto et al., 2014*):

- Play-related behaviors: pouncing (on rat solicits another rat to play by attempting to nose or rub the nape of its neck), pinning (the rat that is pounced upon can respond in different ways – 'pinning' is the result when the animal lies with its dorsal surface on the floor with the other rat standing over it) and boxing (rats push or paw at each other with their forepaws, usually around the head, neck, shoulders of their opponent).
- Social behaviors unrelated to play: social exploration (sniffing any part of the body of the test partner), social grooming (one rat licks and chews the fur of the conspecific, while placing its forepaws on the back or the neck of the other rat), following/chasing (walking or running in the direction of the partner which stays where it is or moves away), crawling under/over (one animal crawls underneath or over the partner's body, crossing it transversely from one side to the other), kicking (the rat kicks backwards at the conspecific with one or both hind paws).
- Aggressive behavior: attack.

## Open field

The test was performed as we previously published (*Manduca et al., 2017*; *Jung et al., 2012*). The apparatus consisted of a Plexiglas arena 45 × 45 cm, illuminated by fluorescent bulbs at a height of 2 m above the floor of the open field apparatus (light intensity of 30 Lux). The floor was cleaned between each trial to avoid olfactory clues. Each animal was transferred to the open-field facing a corner and was allowed to freely explore the experimental area for 10 min. A video tracking system (Ethovision XT, Noldus Information Technology) recorded the exact track of each rat as well as total distance traveled (*Lafourcade et al., 2011*; *Larrieu et al., 2012*).

## Elevated plus-maze

The elevated plus-maze apparatus comprised two open (50 × 10 × 40 cm) and two closed arms (50 × 10 × 40 cm) that extended from a common central platform (10 × 10 cm). The test was performed as previously described (*Manduca et al., 2015*). Rats were individually placed on the

*Figure 10 continued*

n = individual rats. Error bars represent SEM. Scale bar: 10 ms, 0.1 mV.

DOI: https://doi.org/10.7554/eLife.36234.017

central platform of the maze for 5 min. Each 5 min session was recorded with a camera positioned above the apparatus for subsequent behavioral analysis carried out an observer, unaware of animal treatment, using the Observer 3.0 software (Noldus, The Netherlands). The following parameters were analyzed:

- % time spent in the open arms (% Time Open): (seconds spent on the open arms of the maze/ seconds spent on the open +closed arms)×100;
- % open arm entries (% Open Entries): (the number of entries into the open arms of the maze/ number of entries into open +closed arms)×100;

## Temporal order

Animals were habituated to the experimental arena (40 × 40 cm) without objects for 10 min daily for 2 days before testing. This task consisted of two sample phases and one test trial (*Barker et al., 2007*). In each sample phase, rats were allowed to explore two copies of an identical object for a total of 4 min. Different objects were used for sample Phases 1 and 2, with a delay between the sample phases of 1 hr. After 3 hr from sample Phase 2, rats performed the test trial (4 min duration) where a third copy of the objects from sample Phase 1 and a third copy of the objects from sample Phase 2 were used. The positions of the objects in the test and the objects used in sample Phase 1 and sample Phase 2 were counterbalanced between the animals. An intact temporal order memory requested the subjects to spend more time exploring the object from Sample 1 (i.e. the object presented less recently) compared with the object from Sample 2 (i.e. the 'new' object). The discrimination ratio was calculated as the difference in time spent by each animal exploring the object from sample Phase 1 compared with the object from sample Phase 2 divided by the total time spent exploring both objects in the test phase.

## Novel object recognition

After 2 days of 10 min habituation to the experimental arena, rats were exposed to the procedure of an acquisition or sample phase, followed by a preference test after a delay of 30 min (*Campolongo et al., 2012*). In the sample phase, duplicate copies (A1 and A2) of an object were placed near the two corners at either end of one side of the arena (8 cm from each adjacent wall). Rats were placed into the arena facing the center of the opposite wall and allowed a total of 4 min in the arena. At test (4 min duration), animals were placed in the arena, presented with two objects in the same positions: one object (A3) was a third copy of the set of the objects used in the sample phase, and the other object was a novel object (B). The positions of the objects in the test and the objects used as novel or familiar were counterbalanced between the rats. The discrimination ratio was calculated as the difference in time spent by each animal exploring the novel compared with the familiar object divided by the total time spent exploring both objects. Exploration was scored when the animal was observed sniffing or touching the object with the nose and/or forepaws. Sitting on objects was not considered to indicate exploratory behavior.

## Drug treatment

The CB1 cannabinoid receptor antagonist SR141716A [5-(4-chloro-phenyl)−1-(2,4-dichlorophenyl)−4-methyl-N-1-piperidinyl-1H-pyrazole-3-carboxamide] (National Institute of Mental Health, USA), the positive allosteric modulator of mGlu$_5$ receptors CDPPB (3-cyano-N-(1,3-diphenyl-1H-pyrazol-5-yl) benzamide) (National Institute of Mental Health, USA), the vanilloid TRPV1 antagonist capsazepine (N-[2-(4-Chlorophenyl)ethyl]−1,3,4,5-tetrahydro-7,8-dihydroxy-2H-2-benzazepine-2-carbothioamide) (Tocris) and the FAAH inhibitor URB597 (Cyclohexylcarbamic acid 3'-(Aminocarbonyl)-[1,1'-biphenyl]−3-yl ester) (Tocris) were dissolved in 5% Tween 80/5% polyethylene glycol/saline and given intraperitoneally (i.p.). CDPPB (0.75 mg/kg) or its vehicle were administered 30 min before testing; URB597 (0.1 mg/kg) or its vehicle were administered 2 hr before testing; SR141716A was administered 30 min before CDPPB or URB597 at a dose that did not induce effects by itself (1 mg/kg) in adult rats (*Manduca et al., 2015*; *Sciolino et al., 2011*). Capsazepine (5 mg/kg) was injected

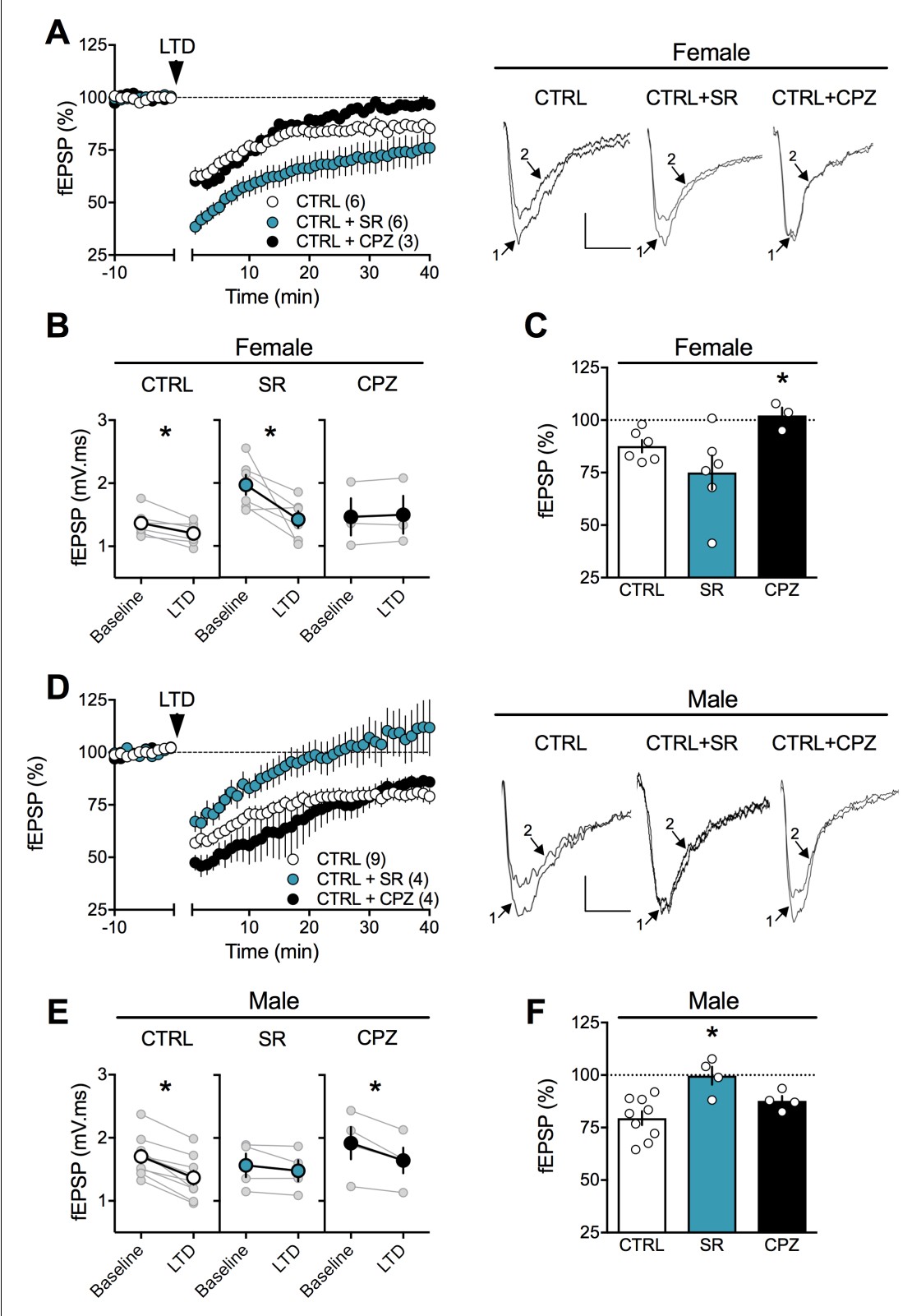

**Figure 11.** Sex-specific substrates of LTD in adult mPFC. (**A**) Time course of normalized field EPSPs recordings from layer 5 mPFC after 10 Hz stimulation protocol (10 min, arrow) in presence of CB1R antagonist SR141716 (SR, n = 6, blue circles) or TRPV1 antagonist Capsazepine (CPZ, n = 3, black circles) in naïve female rats. Right: example traces, baseline (1) and 40 min poststimulation (2)(**B**) Individual experiments (grey) and group average (tan circles), before (baseline) and after (40 min) LTD induction (arrow) showing that, in control female rat (CTRL, left), LTD is blocked by the TRPV1

*Figure 11 continued on next page*

Figure 11 continued

antagonist (CPZ, right) but not by the CB1R antagonist (SR, middle). In control rats: 1.368 ± 0.089 mV.ms before LTD versus 1.205 ± 0.074 after LTD induction (n = 6; p<0.05, paired t-test). After SR141716A preincubation: 1.970 ± 0.160 mV.ms before LTD versus 1.420 ± 0.133 after LTD induction (n = 6; p<0.05, paired t-test). In control female, LTD induction requires activation of TRPV1 receptors: 1.463 ± 0.295 mV.ms before LTD versus 1.497 ± 0.300 after LTD induction (n = 3; p>0.05, paired t-test). (C) Bar chart summary of fEPSP percentage change from baseline 38–40 min after 10 Hz stimulation in control female (white bar) after treatment with CB1R antagonist (SR, blue bar) or TRPV1 antagonist (CPZ, black bar) ($F_{(2, 12)}$=3.928; p<0.05, one-way ANOVA). (D) Time course of normalized field EPSPS recordings from layer 5 mPFC after 10 Hz stimulation protocol (10 min, arrow) in presence of CB1R antagonist SR141716 (SR, n = 4, blue circles) or TRPV1 antagonist Capsazepine (CPZ, n = 4, black circles) in naive male rats. Right: example traces, baseline (1) and 40 min poststimulation (2). (E) Individual experiments (grey) and group average (colors), before (bsl) and after (40 min) LTD induction showing that LTD in control male rats is CB1R-mediated (middle) rather than TRPV1-mediated (right). Low-frequency stimulation of mPFC slices triggers LTD in male (left, white): 1.704 ± 0.107 mV.ms before 10 Hz stimulation versus 1.367 ± 0.112 after LTD induction (n = 9; p<0.05, paired t-test). This LTD is blocked by CB1R antagonist SR141716A: 1.565 ± 0.185 mV.ms before and 1.480 ± 0.167 after LTD induction (n = 4; p>0.05, paired t-test). In contrast, in male rats, LTD is not blocked by TRPV1 antagonist: 1.917 ± 0.256 mV.ms before LTD versus 1.644 ± 0.204 after LTD induction (n = 4; p<0.05, paired t-test). (F) Bar chart summary of fEPSP percentage change from baseline 38–40 min after LTD induction in control male (white bar) after treatment with CB1R antagonist (SR, blue bar) or TRPV1 antagonist (CPZ, black bar) ($F_{(2, 14)}$=7.623; p<0.05, one-way ANOVA). mPFC slices were preincubated for 45 min in 5 µM SR141716A and 10 µM Capsazepine. *p<0.05. n = individual rats. Error bars indicate SEM. Scale bar: 10 ms, 0.1 mV.
DOI: https://doi.org/10.7554/eLife.36234.018
The following figure supplement is available for figure 11:

**Figure supplement 1.** Sex-specific substrates of LTD in naïve adult mPFC.
DOI: https://doi.org/10.7554/eLife.36234.019

30 min before CDPPB and URB597. Drug doses and pre-treatment intervals were based on the literature (*Manduca et al., 2015*; *Ratano et al., 2017*) and on pilot experiments. Solutions were freshly prepared on the day of the experiment and were administered in a volume of 1 ml/kg.

## Physiology
### Slice preparation
Adult male and female rats were anesthetized with isoflurane and killed as previously described (*Martin and Manzoni, 2014*; *Manduca et al., 2017*). The brain was sliced (300 µm) in the coronal plane with a vibratome (Integraslice, Campden Instruments) in a sucrose-based solution at 4°C (in mm as follows: 87 NaCl, 75 sucrose, 25 glucose, 2.5 KCl, 4 MgCl2, 0.5 CaCl$_2$, 23 NaHCO$_3$ and 1.25 NaH$_2$PO$_4$). Immediately after cutting, slices containing the medial prefrontal cortex (PFC) or accumbens were stored for 1 hr at 32°C in a low-calcium ACSF that contained (in mm) as follows: 130 NaCl, 11 glucose, 2.5 KCl, 2.4 MgCl$_2$, 1.2 CaCl$_2$, 23 NaHCO$_3$, 1.2 NaH$_2$PO$_4$, and were equilibrated with 95% O2/5% CO2 and then at room temperature until the time of recording. During the recording, slices were placed in the recording chamber and superfused at 2 ml/min with either low Ca$^{2+}$ ACSF for mPFC or normal ACSF for the accumbens. All experiments were done at 32°C or room temperature for mPFC and accumbens respectively. The superfusion medium contained picrotoxin (100 µM) to block gamma-aminobutyric acid types A (GABA-A) receptors. All drugs were added at the final concentration to the superfusion medium.

### Electrophysiology
Whole cell patch-clamp of visualized layer five pyramidal neurons mPFC and field potential recordings were made in coronal slices containing the mPFC or the accumbens as previously described (*Kasanetz and Manzoni, 2009*; *Kasanetz et al., 2010*; *Kasanetz et al., 2013*). Neurons were visualized using an upright microscope with infrared illumination. The intracellular solution was based on K$^+$ gluconate (in mM: 145 K$^+$ gluconate, 3 NaCl, 1 MgCl$_2$, 1 EGTA, 0.3 CaCl$_2$, 2 Na$_2^+$ATP, and 0.3 Na$^+$ GTP, 0.2 cAMP, buffered with 10 HEPES). To quantify the AMPA/NMDA ratio we used a CH$_3$O$_3$SCs-based solution (in mM: 128 CH$_3$O$_3$SCs, 20 NaCl, 1 MgCl$_2$, 1 EGTA, 0.3 CaCl$_2$, 2 Na$_2^+$-ATP, and 0.3 Na$^+$ GTP, 0.2 cAMP, buffered with 10 HEPES, pH 7.2, osmolarity 290 – 300 mOsm). The pH was adjusted to 7.2 and osmolarity to 290 – 300 mOsm. Electrode resistance was 4 – 6 MOhms.

Whole cell patch-clamp recordings were performed with an Axopatch-200B amplifier as previously described (*Jung et al., 2012*; *Kasanetz and Manzoni, 2009*; *Lafourcade et al., 2011*; *Manduca et al., 2017*; *Martin et al., 2016b*;; *Martin et al., 2016a*; *Robbe et al., 2002a*;

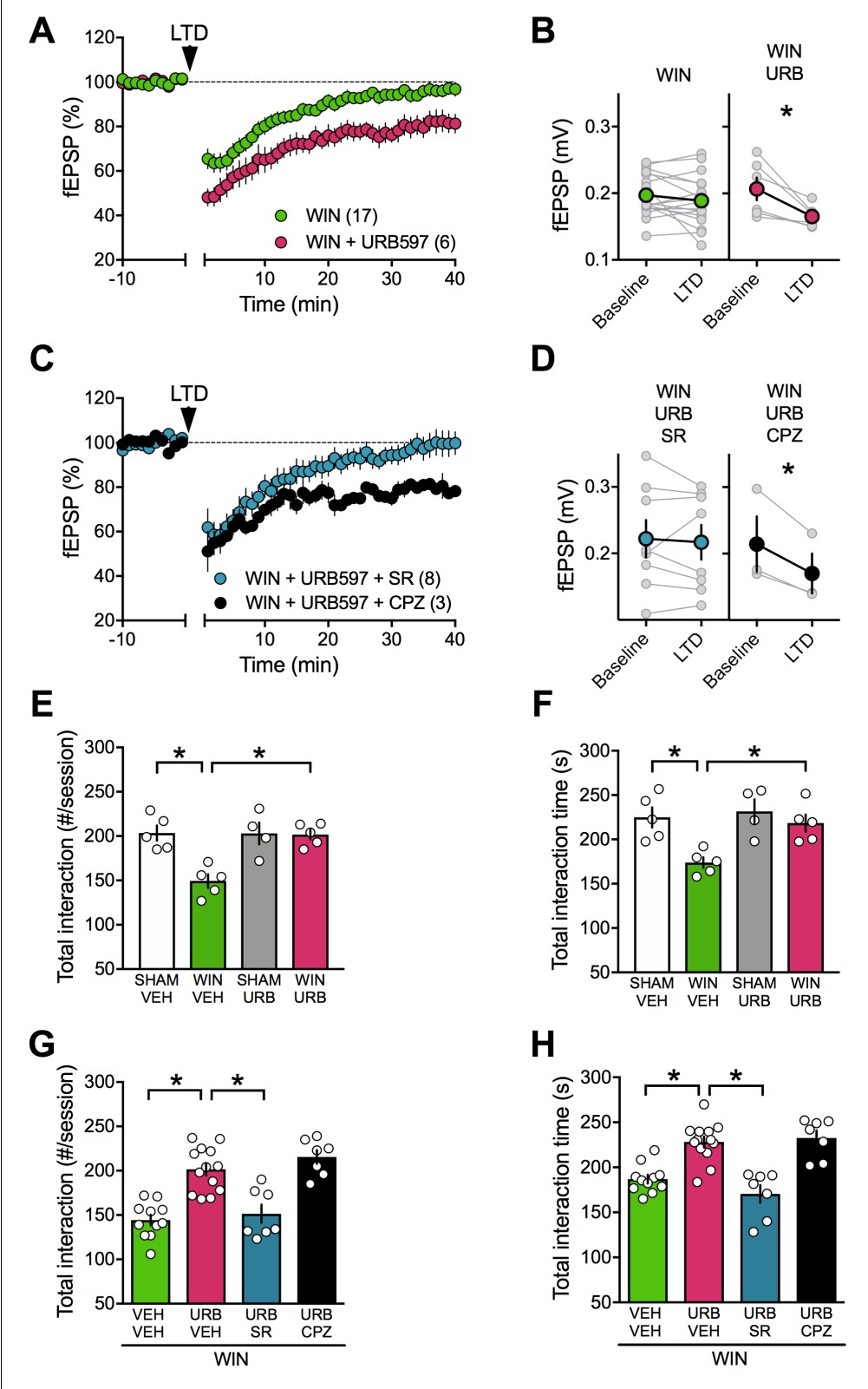

**Figure 12.** FAAH inhibition restores LTD and social interaction in male rats prenatally exposed to cannabinoids. (**A**) Average tie-courses of mean fEPSPs after 10 Hz LTD protocol (1à min, arrow) in WIN-exposed mPFC slices preincubated for 45 min in 2 µM URB597, a FAAH inhibitor. (**B**) Individual experiments (grey) and group averages (tan circles), before (baseline) and after (40 min) LTD induction showing that, in WIN-exposed male (WIN, n = 17, green circle), LTD is restored following bath-application of the FAAH inhibitor URB597 (2 µM, 45 min preincubation, URB, n = 6; pink circle):
*Figure 12 continued on next page*

*Figure 12 continued*

0.207 ± 0.017 mV before and 0.165 ± 0.007 after LTD induction (n = 6; p<0.05, paired t-test). (C) The URB597 LTD rescue requires CB1R. Time course of fEPSP during 10 Hz stimulus evoked LTD (arrow, 10 min) in presence of CB1R antagonist SR141716 (SR, 4 µM, n = 8, blue circles) or TRPV1 antagonist Capsazepine (CPZ, 10 µM, n = 3, black circles) in WIN-prenatally exposed males. (D) Individual experiments: SR141716A preincubation prevented from the URB597 restoration of LTD: 0.222 ± 0.028 mV before and 0.217 ± 0.026 after LTD induction (blue circle, n = 8; p>0.05, paired t-test). In contrast, the TRPV1 antagonist did not prevent from URB597 rescue: 0.214 ± 0.042 mV before LTD and 0.170 ± 0.030 mV after LTD induction (black circle, n = 3; p<0.05, paired t-test). (E–F) Systemic administration of URB597 (0.1 mg/kg, i.p.) normalized the altered social behavior in male rats prenatally exposed to WIN (numbers of contacts: E: $F_{(WIN\ in\ utero\ x\ URB)1,15}$=9.341. p<0.05, two-way ANOVA; time interaction: F: $F_{(WIN\ in\ utero\ x\ URB)1,15}$=3.421, p=0.08, two-way ANOVA). (G-H) Pre-treatment with the CB1R antagonist SR141716A (1 mg/kg, i.p.) prevented the curating actions of URB597 (0.1 mg/kg, i.p.) on social interaction. In contrast, the TRPV1 antagonist capsazepine did not block the curating effects of URB597 on social interaction in adult rats prenatally exposed to WIN55,212–2 (number of contacts: G: $F_{3,34}$=21.02, p<0.05, one-way ANOVA; time interaction: H: $F_{3,34}$=18.34, p<0.05, one-way ANOVA). Scatter dot plot represents each animal. Error bars indicate SEM. *p<0.05 Student–Newman–Keuls test.
DOI: https://doi.org/10.7554/eLife.36234.020

*Robbe et al., 2002b*; *Thomazeau et al., 2014*; *Thomazeau et al., 2017*). Data were low-pass filtered at 2 kHz, digitized (10 kHz, DigiData 1440A, Axon Instrument), collected using Clampex 10.2 and analyzed using Clampfit 10.2 (all from Molecular Device, Sunnyvale, USA).

A −2 mV hyperpolarizing pulse was applied before each evoked EPSC in order to evaluate the access resistance and those experiments in which this parameter changed >25% were rejected. Access resistance compensation was not used, and acceptable access resistance was <30 MOhms. The potential reference of the amplifier was adjusted to zero prior to breaking into the cell. Cells were held at −76 mV.

*Current-voltage (I-V) curves* were made by a series of hyperpolarizing to depolarizing current steps immediately after breaking into the cell. Membrane resistance was estimated from the I–V curve around resting membrane potential (*Kasanetz et al., 2010*; *Martin et al., 2015*).

*Field potential recordings* were made in coronal slices containing the mPFC or the accumbens as previously described (*Kasanetz and Manzoni, 2009*; *Kasanetz et al., 2010*; *Kasanetz et al., 2013*). During the recording, slices were placed in the recording chamber and superfused at 2 ml/min with either low $Ca^{2+}$ ACSF for mPFC or normal ACSF for the accumbens. All experiments were done at 32°C or room temperature for mPFC and accumbens, respectively. The superfusion medium contained picrotoxin (100 µM) to block GABA Type A (GABA-A) receptors. All drugs were added at the

**Table 3.** Reproduction data and pup weight after in utero WIN exposure.
Dam weight gain was calculated from GD 1 to GD 21 for n = 10 dams per treatment group. Pup weight at different postnatal days (PND) was calculated for n = 9–10 male and female pups from different litters. Data represent mean values ± SEM. Statistical significance was determined using the one-way ANOVA test.

| | | Naive | SHAM | WIN | |
| --- | --- | --- | --- | --- | --- |
| | | mean ± SEM | mean ± SEM | mean ± SEM | p-Value (ANOVA) |
| | Dam weight gain (%) | | | | |
| | | 33.8 ± 0.93 | 34.3 ± 1.91 | 33.1 ± 1.42 | 0.745 |
| | Pregnancy length (days) | | | | |
| Reproduction data | | 22.4 ± 0.14 | 22.6 ± 0.29 | 22.6 ± 0.17 | 0.923 |
| | Litter size | 13 ± 0.82 | 12.9 ± 0.78 | 12.7 ± 0.67 | 0.854 |
| | PND 1 | 6.8 ± 0.2 | 6.7 ± 0.1 | 6.8 ± 0.2 | 0.965 |
| | PND 10 | 23.5 ± 0.89 | 24.6 ± 0.65 | 22.6 ± 0.72 | 0.207 |
| Pup weight (grams) | PND 13 | 31.8 ± 0.62 | 30.3 ± 0.82 | 30.5 ± 0.75 | 0.339 |
| | PND 25 | 64.5 ± 2.24 | 65.6 ± 0.77 | 66.7 ± 1.55 | 0.647 |
| | PND 90 | 452.7 ± 12.28 | 484.2 ± 12.52 | 481.7 ± 17.33 | 0.261 |

DOI: https://doi.org/10.7554/eLife.36234.021

final concentration to the superfusion medium. The glutamatergic nature of the field EPSP (fEPSP) was systematically confirmed at the end of the experiments using the ionotropic glutamate receptor antagonist CNQX (20 μM), which specifically blocked the synaptic component without altering the non-synaptic component (data not shown).

Both fEPSP area and amplitude were analyzed. Stimulation was performed with a glass electrode filled with ACSF and the stimulus intensity was adjusted ~60% of maximal intensity after performing an input–output curve (baseline EPSC amplitudes ranged between 50 and 150 pA). Stimulation frequency was set at 0.1 Hz.

### Data acquisition and analysis

The magnitude of plasticity was calculated 35–40 min after induction as percentage of baseline responses. Statistical analysis of data was performed with Prism (GraphPad Software) using tests indicated in the main text after outlier subtraction. All values are given as mean ±SEM, and statistical significance was set at $p < 0.05$.

### Biochemistry *qRT-PCR*

Frozen rat brains were coronally sectioned in a cutting block (Braintree Scientific, Inc., Braintree, MA, Cat.# BS-SS 605C) that had been pre-chilled to −20°C. One-millimeter sections were kept frozen throughout dissection with brain regions stored at −80°C until use. Total RNA was extracted using the RNeasy Plus Micro Kit (Qiagen, Hilden, Germany, Cat.# 74034). RNA was reverse-transcribed using the RevertAid Kit (Thermo Fisher Scientific, Waltham, MA Cat.# K1621) as per manufacturer's instructions. Taqman primers and probes were obtained through Applied Biosystems. Sequences used in qRT-PCR are as follows:

| | Sense | Anti-Sense | Probe | Fluorophore | Quencher |
|---|---|---|---|---|---|
| CB1 | tttcaagcaaggagcaccca | ggtacggaaggtggtgtctg | ctttctcagtccaccttgagtctggcct | FAM | Qsys |
| CB2 | ccacgccgtgcctgagtgag | ccgccattggagccgttggt | cgaggccacccagcaaacatct | FAM | Qsys |
| DAGLalpha | gggtctgaaaccaaacacgc | gacacagtggggagttggag | ctgcccacttgctctcctggc | FAM | Qsys |
| DAGL-beta | tacggatggcccctctacat | gctcgtacaccttgtcgtga | agcaccactgccacttcgcc | FAM | Qsys |
| MAGL | cccttcaggggtgtgttctg | ctttgggccctgtttccattagtc | actcgaggctgtggcggtagt | FAM | Qsys |
| FAAH | gttcaccttggaccctaccg | agaagggaatcagcgtgtgg | accatgcccagcccagctatga | FAM | Qsys |
| NAPE-PLD | ctgcgtgcagctgttactgtc | atgtctttccgtgggaagcttga | caggccctccggtgaggagac | FAM | Qsys |
| TRPV1 | cctagctggttgcaaattggg | tggaggtggcttgcagttag | ccaccccaagagaactcctgcct | FAM | Qsys |
| mGluR5 | agacggcaaatcatcgtccg | ttttccgttggagcttagggtt | tgccagcagatccagcagccta | FAM | Qsys |
| mGluR1 | ggatgctcccggaaggtatg | tcagcactccttcatgccag | tctgcagtacacagaagctaatcgc | FAM | Qsys |
| CRIP1a | aacactgcaggtcgagaaca | tgatctggatgggttgtcgc | tggtgtgcttgtcccactgga | FAM | Qsys |
| ABHD6 | cctcggccactgaggtgtc | gaagctacgaaggccaggat | tgtgattgcgggtgggaccct | FAM | Qsys |
| KCC2 | ccatctacgcaggggtcatc | ggcgggaggaacagaatagg | tttgcctcctggggaaccgc | FAM | Qsys |
| GAPDH | gacttcaacagcaactcccat | tcttgctctcagtatccttgct | agaaaccctggaccacccagcc | Vic | Qsys |

TaqMan Gene Expression Master Mix from Applied Biosystems (Foster City, CA Cat.# 4369016) was used in generating expression data on a QuantStudio7 thermal cycler. Duplicates were run for each sample, and changes in gene expression were determined using the ΔΔCt method. Data was analyzed using Excel (Microsoft, Redmond, WA) and Prism 7 (Graphpad, La Jolla, CA) software.

## Statistics

All the behavioral parameters were expressed as mean ± SEM. Group comparisons used t-test and two-way repeated measures (RM) ANOVA (for different sex and pharmacological treatments), followed by Student-Newman-Keuls post-hoc tests when appropriate.

## Acknowledgements

The authors are grateful to Mr. Clément Bouille for experimental help, Drs. Pascale Chavis and Andrew Scheyer for helpful discussions and to the National Institute of Mental Health's Chemical Synthesis and Drug Supply Program (Rockville, MD, USA) for providing CNQX, CDPPB, URB597 and SR141716A.

## Additional information

### Competing interests

Olivier J Manzoni: Reviewing editor, *eLife*. The other authors declare that no competing interests exist.

### Funding

| Funder | Grant reference number | Author |
|---|---|---|
| Conselho Nacional de Desen-volvimento Científico e Tecno-lógico | | Milene Borsoi |
| Agence Nationale de la Re-cherche | Cannado | Anissa Bara<br>Anne-Laure Pelissier-Alicot<br>Olivier J Manzoni |
| Fondation pour la Recherche Médicale | Equipe FRM 2015 | Anissa Bara<br>Milene Borsoi<br>Olivier J Manzoni |
| National Institutes of Health | 5R01DA043982-02 | Ken Mackie<br>Olivier J Manzoni |

The funders had no role in study design, data collection and interpretation, or the decision to submit the work for publication.

### Author contributions

Anissa Bara, Antonia Manduca, Data curation, Formal analysis, Validation, Investigation, Writing—original draft; Axel Bernabeu, Conceptualization, Data curation, Formal analysis, Validation; Milene Borsoi, Data curation, Formal analysis, Validation, Writing—review and editing; Michela Serviado, Michelle Murphy, Jim Wager-Miller, Data curation, Formal analysis, Methodology; Olivier Lassalle, Data curation, Formal analysis, Validation, Methodology; Ken Mackie, Conceptualization, Supervision, Funding acquisition, Methodology, Writing—review and editing; Anne-Laure Pelissier-Alicot, Conceptualization, Supervision, Funding acquisition, Project administration, Writing—review and editing; Viviana Trezza, Data curation, Formal analysis, Supervision, Validation, Investigation, Methodology, Writing—review and editing; Olivier J Manzoni, Conceptualization, Supervision, Funding acquisition, Methodology, Writing—original draft, Project administration, Writing—review and editing

### Author ORCIDs

Ken Mackie http://orcid.org/0000-0001-8501-6199
Olivier J Manzoni http://orcid.org/0000-0002-5579-6208

### Ethics

Animal experimentation: Animals were treated in compliance with the European Communities Council Directive (86/609/EEC) and the United States National Institutes of Health Guide for the care and use of laboratory animals.

### Decision letter and Author response

Decision letter https://doi.org/10.7554/eLife.36234.024
Author response https://doi.org/10.7554/eLife.36234.025

## Additional files

### Supplementary files

• Transparent reporting form
DOI: https://doi.org/10.7554/eLife.36234.022

### Data availability

The datasets generated during the course of our study will be available upon request to researchers for purposes of reproducing the results or replicating the procedure. Alternatively, processed data can be provided, i.e. numerical data currently represented as a graph in a figure or table.

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
