## [Decision Letter]

Thank you for submitting your article "Sex specific endophenotypes of in-utero cannabinoid exposure" for consideration by *eLife*. Your article has been reviewed by a Senior Editor, a Reviewing Editor, and three reviewers. The following individuals involved in review of your submission have agreed to reveal their identity: Camilla Bellone (Reviewer #2); Carl Lupica (Reviewer #3).

The reviewers have discussed the reviews with one another and the Reviewing Editor has drafted this decision to help you prepare a revised submission. In general, all reviewers were enthusiastic about the work, though revision will be required. In particular, the way social interactions were tested requires further information, and may require more experimental evidence. Specific essential revisions are listed below:

Essential revisions:

1) The social interactions are tested after 24 hrs of isolation; the authors explain that this isolation period is needed in order to enhance social motivation and to facilitate the social behavior they measure. The social isolation per se could modify social interaction differently between WIN treated and SHAM treated rats, and/or between males and females. The authors should report the results without prior isolation to control for these important possibilities that would change the interpretation of the results.

2) The authors use the drug SR141716A to block TRPV1 receptors at 5 μM in the slice experiments. This drug has significant effects at TRPV1 receptors, with a reported Ki of around 1 μM (De Petrocellis et al., 2001). In Figure 9B and D, e.g. in which both this drug and capsazepine prevents rescue of LTD or social interaction by the PAM, it is possible that this effect occurs as a result of interactions with both receptors. This interpretation is made less likely by the fact that differential effects of the two drugs at the same in vitro doses are seen in females, but it may still be worth testing a CB1 selective antagonist instead.

3) The title of the manuscript is a bit cryptic. Maybe something like "Sex-dependent effects of in utero cannabinoid exposure on cortical function" would help sort it out?

---

## [Author Response]

Essential revisions:1) The social interactions are tested after 24 hrs of isolation; the authors explain that this isolation period is needed in order to enhance social motivation and to facilitate the social behavior they measure. The social isolation per se could modify social interaction differently between WIN treated and SHAM treated rats, and/or between males and females. The authors should report the results without prior isolation to control for these important possibilities that would change the interpretation of the results.

Our response is twofold:

1) In our experiments, we tested the animals for social interaction after a brief period of social isolation (24 hours) for the following reasons:

a) In the social interaction test, the animals are normally singly housed for a short period before the test (File and Seth, 2003). This is a standard and widely accepted procedure, since several independent groups have repeatedly shown that social isolation before testing reliably increases the time spent by the animals in social interaction. As originally standardized by File and Hyde in 1978, the social interaction test used 5 days of individual housing, and Niesink and van Ree (1982) showed that the interaction was maximal after 4–7 days of individual housing. They also reported that the increase in social interactions was not due to a general increase in locomotor or exploratory behavior, since no differences in ambulation between individually- and group-housed animals were observed when they were tested together in the social interaction test (Niesink and van Ree, 1982).

b) It should be considered that if rats are group-housed and tested for social interaction, sequential removal of the rats from the cage (cohort removal) may act as a rapid and potent stressor for the remaining rats (File and Seth, 2003). To support this possibility, when rats were treated in triads, those removed last position had shorter social interaction time and higher body temperature than those removed first (Kask et al., 2001a).

c) The deleterious consequences of isolation rearing are usually observed only when animals are isolated for several weeks. However, we agree with the reviewers that even a short isolation period before testing may change the response to drugs. For this reason, we set up a protocol in which animals are isolated for 24 hours only before testing. This period of isolation before testing (that is shorter than in the protocols originally conceived by File and coworkers and van Ree and coworkers and widely accepted worldwide) allows to simultaneously enhance the social motivation of the animals while avoiding cohort removal stress and potential confounding effects due to longer (i.e., several days) isolation housing. Such a protocol has been successfully used in our previous studies.

2) We also performed the experiments suggested by the reviewers. Thus, a new series of experiments was executed in naïve animals. We did not find sex- or social isolation-depending differences in the total events or time of social interaction (Figure 1—figure supplement 1). Specifically, male and female naïve animals not isolated before testing had similar number of contacts and time spent exploring their congeners than male and female naïve animals isolated 24h before testing (Figure 1—figure supplement 1). Detailed exploration of the various parameters of the social session revealed that the frequency and time of sniffing (Figure 1—figure supplement 1C-D) was unchanged; the play events (i.e. pinning, pouncing and boxing) were reduced in female animals (Figure 1—figure supplement 1E), while the number of attack events remained unchanged (Figure 1—figure supplement 1F).

In conclusion, although we cannot exclude that in-utero treated animals the 24h isolation period could modify social interaction differently than in naïve animals, the data (response 2 above) and the available literature (response 1 above) do not support this hypothesis.

*2) The authors use the drug SR141716A to block TRPV1 receptors at 5 μM in the slice experiments. This drug has significant effects at TRPV1 receptors, with a reported Ki of around 1 μM (De Petrocellis et al., 2001). In Figure 9B and D, e.g. in which both this drug and capsazepine prevents rescue of LTD or social interaction by the PAM, it is possible that this effect occurs as a result of interactions with both receptors. This interpretation is made less likely by the fact that differential effects of the two drugs at the same* in vitro *doses are seen in females, but it may still be worth testing a CB1 selective antagonist instead.*

The reviews rightfully raise the important point of SR141716A’ selectivity for CB1R. One could argue that the De Petrocellis’ study, performed in human VR1-overexpressing HEK cells is hard to translate to rat brain slices. However, the paper by Gibson et al., (2008) convincingly shows how in mice hippocampus SR141716A (2µM) can inhibit CB1R and TRPV1R while AM251 (2µM) is more selective for CB1R. As pointed out by the reviewers, taken as a whole, our experiments strongly argue against the possibility that SR141617A blocks prefrontal synaptic plasticity via TRPV1R. Differences in the species (rat vs mice), neuronal types (GABA interneuron, Glutamatergic pyramidal neurons) and brain areas (hippocampus vs prefrontal cortex) may also account for pharmacological differences.

Thus, beside the CDPPB experiment criticized by the reviewers, the rest of our data clearly show differential effects of SR141716A and capsazepine (a TRPV1R antagonist).

While we remain convinced that prenatal cannabis has long-term sex specific effects on behavior and synaptic properties, we decided to address experimentally the concerns raised by the reviewers. We have performed a series of new experiments in naïve animals with either the selective CB1 antagonist NESS0327 (1µM, Meye et al., 2013; Lee et al., 2015) and the selective TRPV1R antagonist AMG9810 (3µM, Gavva et al., 2005). As expected from our previous experiments with SR and Capsazepine, the CB1R antagonist NESS0327 prevented LTD induction in naïve rat male but not female slices (Figure 11—figure supplement 1) as observed with SR141716A treatment. In contrast, antagonism of TRPV1R with AMG9810 was without effect in males but prevented the induction of eCB-LTD in naïve females (Figure 11—figure supplement 1).

The new data confirm that CB1R (not TRPV1R) mediate LTD in adult male while TRPV1R (not CB1R) mediate LTD in adult female rat PFC. They also show that, in our model (adult rat PFC) SR does not seem to antagonize TRPV1R.

Based on these new data fully displayed in the revised manuscript, we feel that controlling for the selectivity of SR in CDPPB-treated prenatally WIN-exposed male group rather unnecessary.

3) The title of the manuscript is a bit cryptic. Maybe something like "Sex-dependent effects of in utero cannabinoid exposure on cortical function" would help sort it out?

The title has been modified to that suggested by the reviewers.